# Networks of geometrically coherent faults accommodate Alpine tectonic inversion offshore southwestern Iberia

**Tiago M. Alves**

3D Seismic Lab, School of Earth and Environmental Sciences, Cardiff University, Main Building, Park Place, Cardiff, CF10 3AT, United Kingdom

**Correspondence:** Tiago M. Alves (alvest@cardiff.ac.uk)

**Abstract.** The structural styles and magnitudes of Alpine tectonic inversion are reviewed for the Atlantic margin of southwestern (SW) Iberia, a region known for its historical earthquakes, tsunamis and associated geohazards. Reprocessed, high-quality 2D seismic data provide new images of tectonic faults, which were mapped to a depth exceeding 10 km for the first time. A total of 26 of these faults comprise syn-rift structures accommodating vertical uplift and horizontal advection (shortening) during Alpine tectonics. At the regional scale, tectonic reactivation has been marked by (a) the exhumation of parts of the present-day continental shelf, (b) local folding and thrusting of strata at the foot of the continental slope, and (c) oversteepening of syn- and post-rift sequences near reactivated faults (e.g. "passive uplift"). This work proves, for the first time, that geometric coherence dominated the growth and linkage of the 26 offshore faults mapped in SW Iberia; therefore, they are prone to reactivate as a kinematically coherent fault network. They form 100–250 km long structures, the longest of which may generate earthquakes with a momentum magnitude ($M_{\mathrm{w}}$) of 8.0. Tectonic inversion started in the Late Cretaceous, and its magnitude is greater close to where magmatic intrusions are identified. In contrast to previous models, this work postulates that regions in which Late Mesozoic magmatism was more intense comprise thickened, harder crust and form lateral buttresses to northwest–southeast compression. It shows these structural buttresses to have promoted the development of early stage fold-and-thrust belts – typical of convergent margins – in two distinct sectors.

## 1 Introduction

Continental margins record complex post-rift tectonic histories, with fault reactivation and associated uplift being often controlled by structures inherited from their syn-rift evolution stages (Vasconcelos et al., 2019; Rodríguez-Salgado et al., 2020; Schiffer et al., 2020). Such a complexity is amplified when syn- and post-rift magmatism combine with far-field tectonics to affect distal offshore regions (Sun et al., 2020; Jolivet et al., 2021). For instance, local uplift and exhumation of older rocks on the Brazilian and West African margins were driven by important magmatism near the Walvis Ridge, Vitória-Trindade Seamount Chain, Pernambuco Plateau, and Fernando de Noronha Seamounts, as key examples, in episodes that spanned several million years (Comin-Chiaramonti et al., 2011; Strganac et al., 2014; Teboul et al., 2017). In the South China Sea, evidence exists of important syn-breakup volcanism, which was followed by post-rift magmatism near basin-bounding faults (Lei et al., 2020). Once again, these phenomena occurred over a time span of ca. 32 Myr (Zhao et al., 2016; Sun et al., 2022). All these new data stress a paradox in the published literature; while previous work tends to link magmatic processes to well-dated episodes, there is now increasing evidence that long-lasting tectonics is a crucial factor controlling the post-rift evolution of continental margins (Duarte et al., 2013; Casson et al., 2021).

In western Iberia, outcropping igneous rocks and offshore magmatic edifices have been correlated with distinct episodes of magmatism (Miranda et al., 2009; Pereira et al., 2022; Neres et al., 2023b). Following its full separation from North America and Eurasia, Iberia's Alpine-related evolution recorded anticlockwise rotation and subsequent

collision with Eurasia (Pyrenean phase) and North Africa (Betic phase) (Vissers et al., 2016; Jolivet et al., 2021). Alkaline magmatism marked the first stages of collision between Iberia and Eurasia, near the Pyrenees (Geldmacher et al., 2016; Martín-Chivelet et al., 2019), and has been associated with strike-slip tectonics and hotspot magmatism in western Iberia (Miranda et al., 2009). In central and southwestern (SW) Iberia, magmatism lasted for 30–40 Ma. However, an aspect not fully addressed in the literature concerns how this post-rift magmatism relates to the modern structural framework of its continental margin (Pereira et al., 2022; Neres et al., 2023a). A deeper knowledge of the links between this post-rift magmatism and the structural framework of Iberia's Atlantic Margin is crucial to understand its full tectonic and thermal evolutions, particularly on its more tectonically active and seismogenic southwestern margin (Fig. 1).

A second aspect not fully addressed concerns the stratigraphic record of Alpine tectonics in SW Iberia, as it is dominated by its younger Miocene pulse (Betic phase). Older Paleogene strata sampled onshore and in exploration wells are relatively thin and too sparse to provide a complete record of tectonic movement. Against this background, Maldonado et al. (1999) and Alves et al. (2003) used seismic and stratigraphic information to recognize Cenozoic phases of extensional collapse offshore western and southern Iberia based on the presence of reactivated syn-rift faults which have caused, during the Late Cenozoic, significant level differences between shallow continental-shelf regions and their immediate continental-slope basins. Such an interpretation contrasts with most seismic data from SW and central Portugal, where evidence for tectonic compression and widespread reactivation of syn-rift structures is observed (Gràcia et al., 2003; Terrinha et al., 2003; Neves et al., 2009; Duarte, 2013). Near the Nubian–Iberian plate boundary, located in the Gulf of Cádiz between Spain and Morocco, multiple tectonic structures have also been mapped by Ramos et al. (2017) and related to Neogene inversion of a Jurassic oblique passive margin previously developing between the Central Atlantic and the Ligurian Tethys. In the particular case of the Atlantic margin of SW Iberia, Terrinha et al. (2003) suggested the presence of linked fault strands, none capable of generating the $M_\mathrm{w}$ 9.0 Lisbon earthquake of 1755, but that were still long enough to be the loci of relatively large, proximal earthquakes. Terrinha et al. (2003) also suggested the presence of a ramp–flat structure at a depth of 6–8 km below the seafloor, with this structure being extensive enough to justify the combined reactivation of two of the largest faults on SW Iberia's continental slope (TTR-10 and PSF). However, the interpretations in Terrinha et al. (2003) and more recent work were not accompanied by a detailed mapping of all major faults crossing SW Iberia using a comprehensive seismic-reflection data set.

Essential to proving a structural link between active seismogenic faults is the recognition of their geometric and kinematic coherence (Walsh and Watterson, 1991; Walsh et al., 2003; Kim and Sanderson, 2005; Fossen and Rotevatn, 2016). Geometric coherence is defined as the development of regular and systematic displacement patterns in a family of faults (Walsh and Watterson, 1991). In parallel, kinematic coherence reflects the existence of synchronous slip rates and slip distributions that are arranged in a way such that geometric coherence is maintained (Peacock et al., 2002). These two types of coherence can occur for any fault types in nature: normal, strike-slip, or reverse faults (Willemse et al., 1997; Davis et al., 2005; Song et al., 2020).

While kinematic coherence is better established by documenting surface deformation after large earthquakes (Sachpazi et al., 2003; Elias and Briole, 2018; Karabulut et al., 2023), geometric coherence in seismic and outcrop data suggests strain in particular structures to be accommodated – at the geological timescale – as a continuum (Walsh et al., 2003). In other words, coherent sets of faults are able to accommodate strain by interacting in time and space, merging at depth to form continuous fault zones, i.e. fault displacement and growth are accommodated at the surface by discrete faults, but these same structures are linked as a continuous fault at depth (Giba et al., 2012). Importantly, geometric and kinematic coherence occur along a series of interacting fault strands when one considers the lateral growth of a fault zone, as exemplified in Giba et al. (2012), but can also span a vast area and multiple strands when successive faults are kinematically related and linked. The best example of this areal span in geometric coherence is recorded by fold-and-thrust belts of accretionary prisms, which form complex fault networks posed to be reactivated in tandem during major seismogenic events, linked at depth by a common basal plate boundary thrust (Tsuji et al., 2014; Kimura et al., 2018). The recent earthquake in Turkey, on 6 February 2023, is another example of the effect of kinematically linked fault segments – in this case part of a long fault zone, the East Anatolian Fault Zone (EAFZ) – when reactivated in sudden, unexpected events (Karabulut et al., 2023). The latter authors stress, in their work, that the EAFZ is an important reminder that large faults can generate large earthquakes in multiple, kinematically linked segments, whose historical record of past seismicity is poorly documented.

This work goes beyond the published data to reveal that geometric coherence typifies the structural style of reactivated faults in SW Iberia. A total of 26 faults have responded to NW–SE compression during the Late Cenozoic, a tectonic setting still active at present (Ribeiro et al., 1996; Somoza et al., 2021), by linking laterally and growing in tandem. Hence, tectonic uplift and horizontal advection (shortening) in SW Iberia are accommodated by sets of faults that reveal a coherent growth mode. Uplift and horizontal advection are also shown to be greater oceanwards from Late Mesozoic magmatic complexes. This interpretation has important implications for future geohazard assessments and to estimates of SW Iberia's seismogenic and tsunamigenic potential. In

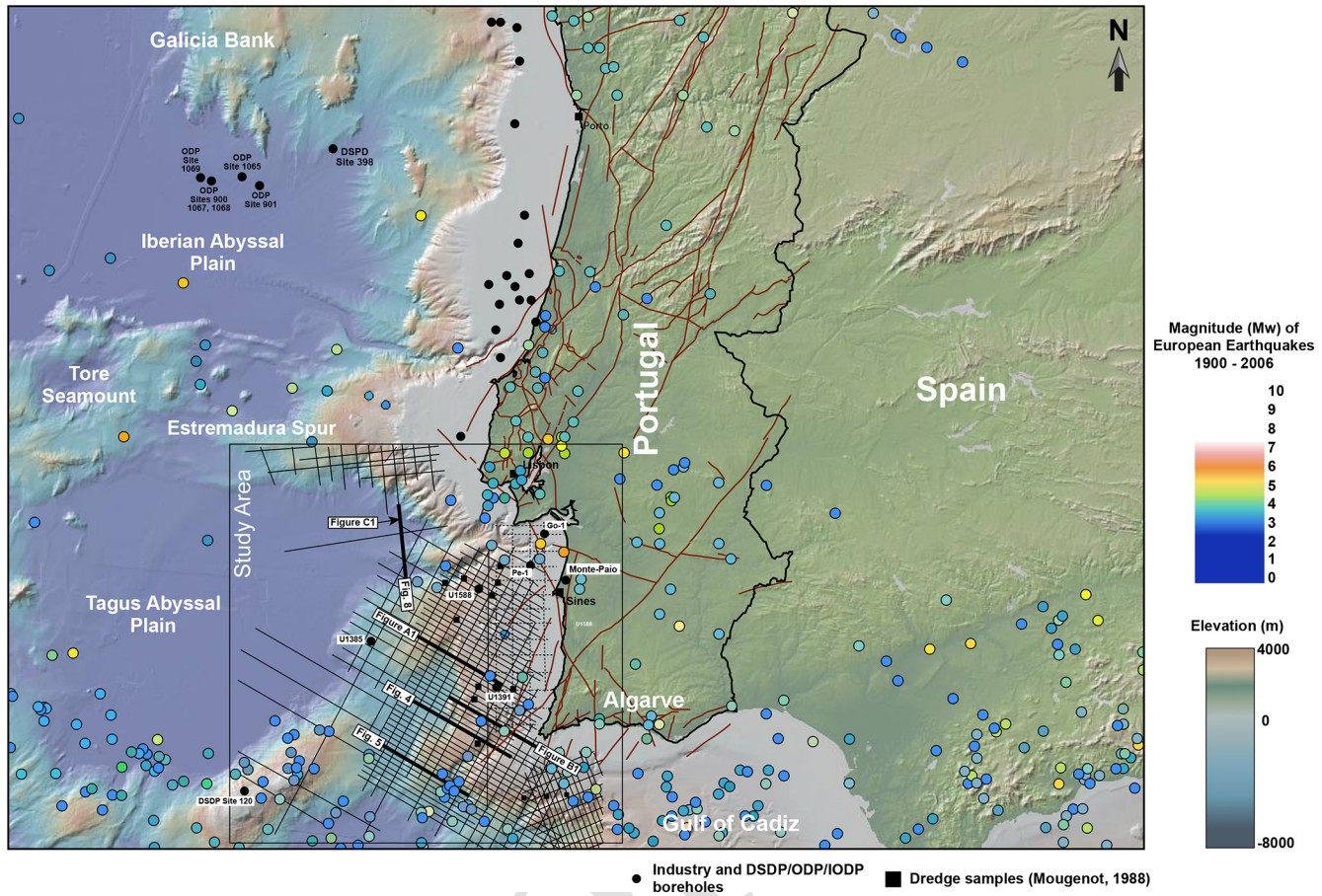

**Figure 1. (a)** Bathymetric and topographic map from western Iberia highlighting the seismic and borehole data sets interpreted in this work, as well as the main physiographic features and faults. The study area is shown within a black box together with the data set interpreted in this work. Also shown on the map are the locations and relative magnitudes of earthquakes for the period spanning 1900 to 2006, as obtained from GeoMapApp (http://www.geomapapp.org, last access: 31 July 2023). The location and extent of onshore faults are taken from Cunha (2019).

summary, this paper addresses the following research questions.

a. How can one quantify the magnitude of tectonic uplift and exhumation in proximal parts of reactivated continental margins using seismic reflection data?

b. In what ways do tectonic uplift and exhumation relate to early magmatism along and across continental margins?

c. Which faults in SW Iberia record the greatest magnitudes of tectonic uplift and horizontal advection, and how do they interact in time and space?

## 2 Geological setting

### 2.1 Basement tectono-magmatic evolution

Basement units on the continental margin of western Iberia comprise a set of Variscan terrains whose limits and nature

are yet not fully understood (Amigo Marx et al., 2022). Tectonic and geophysical data suggest the deep-offshore basins of western Iberia are underlain by a tectonic terrain not identified onshore (Ribeiro et al., 2013), while alternative interpretations suggest these onshore Variscan terrains to extend oceanwards under offshore sedimentary basins imaged in seismic data, thus complying with the general NW-to-ESE strikes of main faults and depocentres recognized in and around the Lusitanian Basin (Capdevilla and Mougenot, 1988). Recent apatite fission-track analyses undertaken by Dinis et al. (2021) have shown sediment in the Lusitanian Basin to comprise a mix of lithological fragments with late Cryogenian–Ediacaran (Pan-African and/or Cadomian, with peaks at 608–554 Ma) and Carboniferous–Permian (Variscan and post-Variscan, with peaks at 315–292 Ma) ages. Some of these fragments are derived from both easterly and westerly sediment sources (see also Walker et al., 2021), showing evidence for having been recycled from eroded sediment. Significantly, some of the samples analysed in Dinis et al. (2021) reflect the presence of basement lithologies that are differ-

ent from the Variscan units now outcropping in Portugal and Spain.

## 2.2  Syn-rift evolution of western Iberia

The western Iberian margin experienced continental rifting from at least the Late Triassic to the Early Cretaceous, preceding a continental breakup phase that spans the latest Jurassic (Tithonian) to the Albian/Cenomanian (Alves and Cunha, 2018). Continental rifting was first widespread on the margin, with progressive lithospheric stretching and thinning resulting in a continental breakup event that propagated from SW to N Iberia, towards what is now the Bay of Biscay (Grevemeyer et al., 2022). In the specific case of SW Iberia, magmatism accompanied syn-rift tectonics in two main episodes: (a) one at ca. 200 Ma (Hettangian) associated with the Central Atlantic Magmatic Province (CAMP) and essentially tholeiitic in nature and (b) a second episode dated from 135 to 130 Ma (Valanginian) with a transitional affinity (Martins et al., 2008).

Continental breakup first started in what is now the Seine and Tagus Abyssal Plains by the latest Jurassic–earliest Berriasian and propagated along western Iberia, in a northerly direction, during the Early Cretaceous (Tucholke et al., 2007; Alves et al., 2009; Neres et al., 2023a). Doubts still exist on the absolute timings of full, established breakup between western Iberia and Canada, though two important details have now been corroborated in the published literature: (a) the *J* anomaly is diachronous and reflects important magmatism associated with the northward propagation of continental breakup (Grevemeyer et al., 2022), (b) the regional stratigraphy records two distinct tectonic pulses of continental breakup – one Berriasian–Aptian(?) associated with breakup offshore SW and central Portugal and a Late Aptian–Cenomanian pulse associated with fully established breakup in NW Iberia (Alves and Cunha, 2018). Recent data from Grevemeyer et al. (2022) and Saspiturry et al. (2021) indicate that Late Aptian–Cenomanian tectonics marks the onset of lithospheric breakup west of Galicia into the Bay of Biscay.

At present, the continental slope of SW Iberia dips gently to the west due to the accumulation of thick Cretaceous–Cenozoic strata (Alves et al., 2009). However, important Late Cretaceous–Cenozoic exhumation and erosion are recorded on its proximal part, where the effect of Alpine tectonics and resulting convergence with Africa were and are still significantly felt (Terrinha et al., 2003; Pereira et al., 2013). Furthermore, a major bathymetric feature – the so-called Estremadura Spur – separates SW Iberia from its NW part and was the locus of important post-rift tectonics and magmatism (Miranda et al., 2009) (Fig. 1). The evolution of the Estremadura Spur is associated with significant compressional tectonics and tectonic inversion in a style akin to a regional-scale "pop-up" structure (Ribeiro et al., 1990). This large pop-up structure trends roughly east–west and is linked to the

onshore, NE–SW-striking, Central Iberian Range (Cunha, 2019).

## 2.3  Post-rift evolution

Post-rift tectonics started with the anticlockwise rotation of the Iberian Plate after continental breakup near the Pyrenees and led to the reactivation of older syn-rift structures (Saspiturry et al., 2020). After 80 Ma (Campanian), important magmatism occurred throughout Iberia (Martín-Chivelet et al., 2019). In SW Iberia, onshore and offshore alkaline magmatism is marked by the presence of sub-volcanic complexes near Sintra, Sines and Monchique (Figs. 1 and 2). Volcanic complexes also occur near Lisbon and offshore Algarve (Miranda et al., 2009). Onshore, this Upper Cretaceous magmatism has been considered to range from ∼ 94 to 69 Ma (Miranda et al., 2009; Grange et al., 2010). Also recorded at this time was the cessation of volcanism in the Basque Basin (Castañares and Robles, 2004), while the Catalan Coastal Ranges recorded the intrusion of isolated, alkaline lamprophyres (Martín-Chivelet et al., 2019). The age of this magmatism is well constrained to ∼ 79 Ma and is considered to mark the onset of Alpine shortening in the easternmost Pyrenean sector (Ubide et al., 2014).

Stratigraphically, Cenozoic tectonic reactivation and uplift are only partly expressed onshore, although important information can be gathered from the Lower Tagus and Alvalade Basins (Fig. 2). Reis et al. (2001) and Cunha (2019) correlate the Benfica Formation to the end of Pyrenean orogenesis and suggest an upper Eocene–Oligocene age for the continental strata forming this unit (Fig. 2). The thin Paleogene strata outcropping near Lisbon are overlain by a significant thickness of Miocene siliciclastics, which are associated in the literature with the Betic orogeny of southern Iberia (Fig. 2). Collision of the African plate with Iberia resulted in the subduction of oceanic crust near Gibraltar, initiating the orogenic episode that generated the Rif and Betic mountain ranges (Zitellini et al., 2009; Gutscher et al., 2012; Monna et al., 2015).

Onshore, stratigraphic information points to a principal middle to late Miocene (Burdigalian to Tortonian) episode of deformation near Lisbon (Arrábida Range), while the Algarve Basin to the south, i.e. closer to the Rif and Betic mountain ranges, documents several stages of Miocene compression (Mougenot, 1988; Cunha et al., 2019) (Fig. 1). Ramos et al. (2016, 2017) have shown inversion in the Algarve Basin to extend beyond the Miocene. Generalized uplift of western Iberia's coastline is also recognized in the Pliocene–Quaternary and locally during major seismo-tectonic events such as the 1755 Lisbon earthquake, which uplifted the Atlantic coast of Iberia in several locations (Silva et al., 2023). Neotectonic activity is also obvious near reactivated structures such as the Pereira de Sousa Fault, the São Vicente Fault, and in multiple areas offshore Algarve (Terrinha et al., 2009; Somoza et al., 2021).

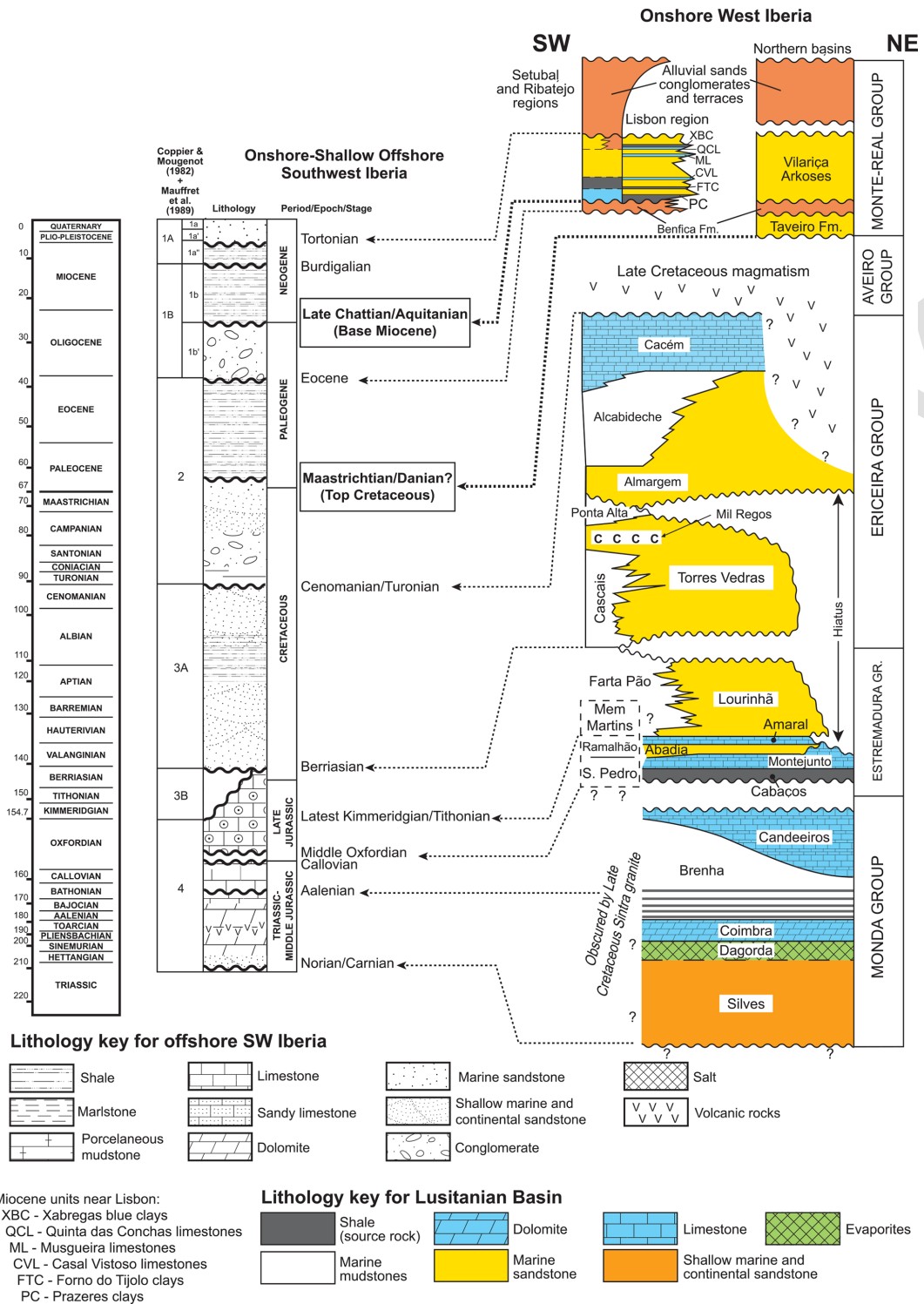

**Figure 2.** Stratigraphic panel correlating main seismic–stratigraphic units offshore SW Iberia with the known stratigraphy of the Lusitanian, Lower Tagus and Alvalade basins (onshore western Iberia). Major seismic–stratigraphic markers in our interpretation include the Base Miocene and Top Cretaceous unconformities, as highlighted in the figure.

## 3 Data and methods

### 3.1 Seismic-well correlations

This work uses regional (2-D) seismic data tied to exploration wells from SW Iberia, as shown in Fig. 1. The interpretation criteria of Alves et al. (2009) and Pereira et al. (2013) are used to map and recognize main seismic–stratigraphic units, reactivated syn-rift structures and associated magmatic edifices. Unpublished information from exploration wells Pe-1, Go-1 and Monte Paio-1, together with dredge data published in Mougenot et al. (1979) and Mougenot (1988), are used to date the main seismic–stratigraphic markers and strata (Fig. 1). These data are complemented with information from DSDP Site 120 and IODP sites U1385, U1391 and U1588, which recently drilled the SW Iberian margin (Hernández-Molina et al., 2013; Hodell et al., 2023) (Fig. 1).

All data are fully integrated in a Schlumberger's Petrel® project so that structural, magnetic and seismic stratigraphic data can be analysed together. The main seismic–stratigraphic markers are interpreted across the study area and, whenever possible, corroborated with information from DSDP and IODP sites U1385, U1391 and U1588 (Fig. 2). A $V_p$ ($p$ wave) velocity of $2000\,\mathrm{m\,s^{-1}}$ was used when estimating fault uplift, and thus translating the values (in ms) measured in time to their corresponding value in metres. This value was gathered as an average $V_p$ value for Cenozoic strata in wells Go-1 and Pe-1 (Fig. 1).

### 3.2 Kinematic data revealing tectonic uplift and horizontal advection

An important aspect of this study concerns the mapping and quantification of tectonic uplift, horizontal advection and folding. Hence, the quantification of tectonic uplift and horizontal advection is based on the criteria illustrated in Fig. 3 (He et al., 2021), together with the recognition of major depositional hiatuses along tectonically uplifted areas of the SW Iberia margin, i.e. the erosion or non-deposition of Late Mesozoic–Cenozoic megasequences that are, on the continental slope and rise, well developed and oversteepened.

The kinematic models in Willet et al. (2001), Willett and Brandon (2002), and more recently He et al. (2021) recognize a significant difference between convergent and extensional regions in terms of their inherent deformation styles (Fig. 3). Contraction of the upper crust will cause the strata to fold or oversteepen, maintaining the bed-parallel geometries of strata that preceded such a contraction (Fig. 3). In other words, tectonic contraction will oversteepen the hanging-wall strata of a thrust (or fold) in its direction of vergence without imposing thickness variations (growth or erosion) to older strata deformed below the seafloor (Fig. 3). Key stratigraphic markers that precede the deformation phase will be tilted and deformed but without revealing syn-kinematic strata growth. In contrast, regions experiencing extension will deform to accommodate vertical subsidence on their hanging-wall blocks and uplift on footwall blocks, with the difference in level between these two blocks leading to important growth of strata in syn-tectonic hanging-wall basins (Fig. 3).

It is thus important to stress that the growth of strata accompanies subsidence in extensional settings, while folding and thrusting is expected in areas experiencing contraction, together with tectonic uplift. If one has reliable stratigraphic markers that were repeatedly oversteepened on a continental slope – though originally laid in a near-horizontal position on a continental margin – a minimum value for uplift and exhumation can be estimated taking into account the paleotopography of a continental margin.

In summary, by mapping throws, dips and level differences in key seismic–stratigraphic markers, one can estimate the minimum tectonic uplift and horizontal advection recorded by particular structures after their syn-rift (extensional) stage. Such a method is akin to the collection of the throw–depth ($T-Z$) and throw–distance ($T-D$) data necessary to characterize the modes of fault growth in normal fault arrays (Walsh and Watterson, 1991; Walsh et al., 2003) but using regional stratigraphic markers of known approximate age as reference. The two key seismic–stratigraphic markers considered in this study consist of the Top Cretaceous and Base Miocene unconformities, as also identified in Gràcia et al. (2003), Terrinha et al. (2003), Alves et al. (2009), and Terrinha et al. (2009). Fault uplift and horizontal advection in this work represent the maximum recorded values recorded between these two markers.

## 4 Reactivated offshore structures

Figures 4, 5 and 6 depict the main structures mapped offshore SW Iberia. A series of NE–SW normal faults compose the structural framework of the margin and often interact spatially to form a mosaic of sub-basins and minor depocentres (Fig. 7). High-amplitude folds and pervasive faulting of Mesozoic and early Cenozoic strata are observed in seismic data. Well-developed, reactivated tilt blocks are recognized to the west, oceanwards of Fault 3 – the Slope Fault System (SFS) of Alves et al. (2009) and the Pereira de Souza Fault (PSF) of Terrinha et al. (2003) (Figs. 4 and 5). Importantly, the region east (landwards) of Fault 3 reveals relatively thin and exhumed Mesozoic rocks (Fig. 4). This character accompanied the oversteepening of post-rift strata on the continental slope, where erosion and exhumation of basement rocks are also observed east of Fault 3 (Figs. 4, A1 and B1).

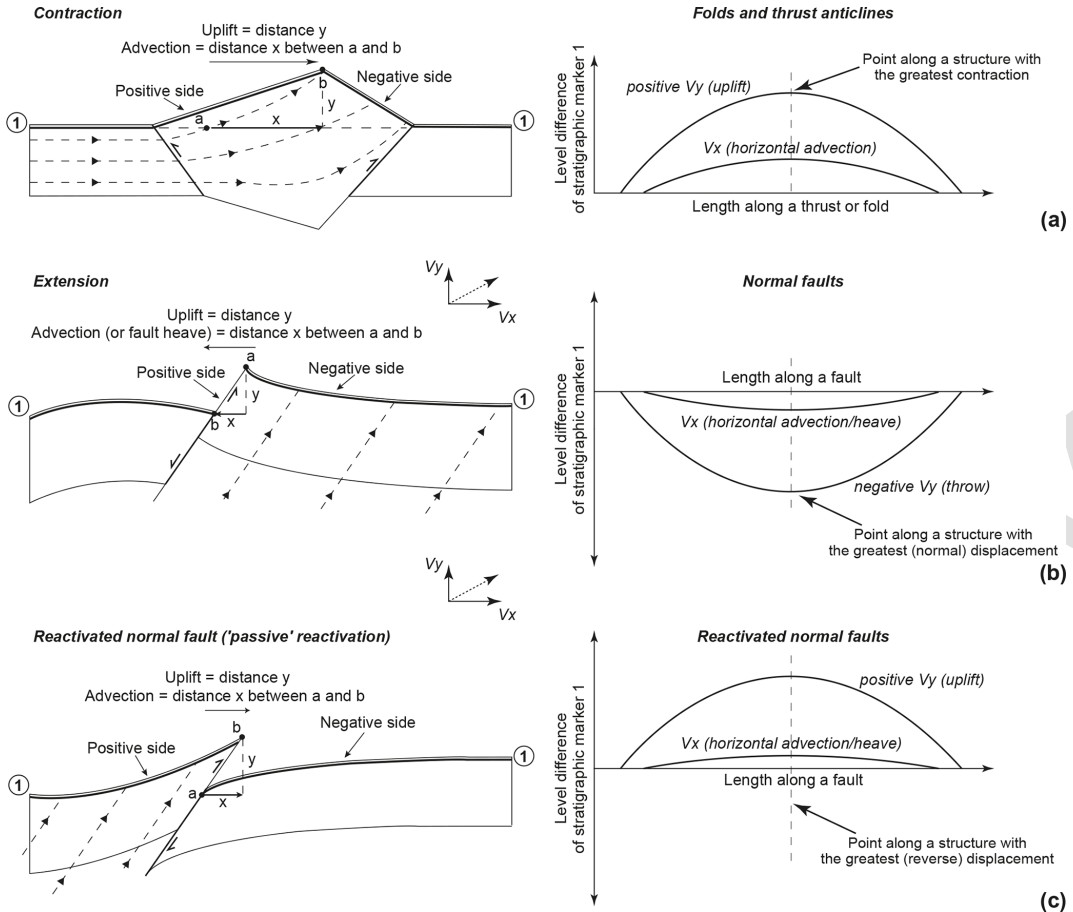

**Figure 3.** Kinematic models for tectonic uplift and horizontal advection as modified from He et al. (2021). Material transport in each of these structures is marked as dashed lines with arrows. Material transport in the three illustrated cases has vertical ($V_x$) and horizontal components ($V_y$). In all configurations, advection is from the positive side to the negative side. **(a)** Uplift and horizontal advection in folds and thrust anticlines, with the level difference of the stratigraphic marker indicating a positive variation for both parameters. **(b)** Uplift and horizontal advection in an extensional setting, with horizontal advection and uplift being negative (throw). **(c)** Uplift and horizontal advection for the case of a normal fault reactivated under a compressive setting, with both uplift and horizontal advection being positive in value. In the normal fault case in **(b)**, an uplift and horizontal advection of zero (0) are assumed in this work and in all measurements undertaken for the sake of simplicity.

## 5 Kinematic indicators of tectonic uplift

### 5.1 Syn-rift fault strands reactivated and laterally linked as reverse faults, thrusts and thrust anticlines

In SW Iberia, the most frequently observed style of tectonic reactivation relates to the inversion of syn-rift normal faults, which were mostly west-dipping faults during the Mesozoic, as east-dipping thrust and reverse faults. Figures 4 and 5 show examples of such reactivation, downslope from the shelf edge and near Fault 7 (the Marquês de Pombal Fault; cf. Terrinha et al., 2003). Slope strata are deformed, oversteepened and locally thrusted, contrasting with the style observed east of Fault 3 (Pereira de Sousa Fault; Terrinha et al., 2003). In the particular case of Fault 7, the seismic data show evidence for the rooting of thrusts and reverse faults on syn-rift horsts, which are offset at depth (Fig. 5). Further north, the past imposition of a rough N–S direction of compression in the Estremadura Spur has reactivated previous syn-rift faults as reverse faults and related anticlines, particularly near the edges of the Spur, forming marked bathymetric features (Figs. 6, 7 and 8). Other anticlines and local popup structures occur throughout the Estremadura Spur, some of which are related to the presence of buried igneous intrusions.

### 5.2 Asymmetrically uplifted (and exhumed) strata on the footwall of thrust and reverse faults

Deformed and oversteepened strata on the footwall of thrust (and reverse) faults are another diagnostic feature of tectonic

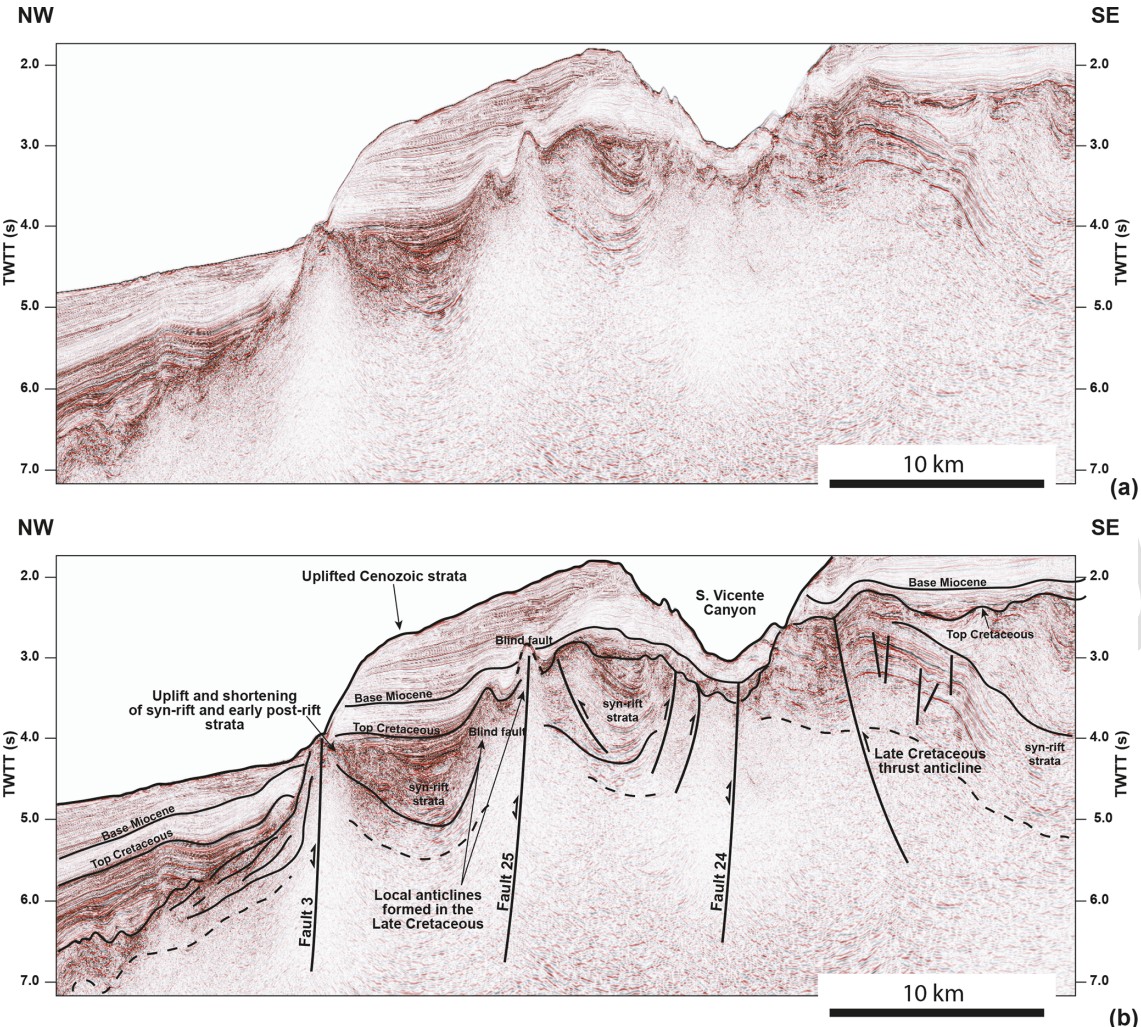

**Figure 4. (a)** Uninterpreted and **(b)** interpreted seismic profile from SW Iberia highlighting the presence of a series of syn-rift faults passively reactivated with a magnitude that was responsible for uplift and tilting of large portions of the continental slope. The profile is shown with a 6× vertical exaggeration. Key seismic stratigraphic markers and units are also indicated in the figure, as well as the location of the S. Vicente Canyon. Note the degree of tectonic uplift recorded to landwards of Fault 3 (i.e. towards the SE) and the marked Late Cretaceous erosion and folding that is imaged below the Top Cretaceous unconformity. The location of the seismic profile is shown in Fig. 1. Seismic data are courtesy of TGS.

uplift and compression (Figs. 4 and B1). During syn-rift extension, these strata were east-dipping, accumulated within relatively flat depocentres reflecting sediment-filled conditions, and developed stratal growth onto faults located to their east. In contrast to their Mesozoic geometry, they are now oversteepened and tilted to the west (Figs. 4 and B1). Many of these oversteepened, uplifted strata not only terminate against Fault 3, which has previously been interpreted as part of a major slope bordering fault system (SFS, Alves et al., 2009) but also onto other reactivated faults and horsts on the continental slope.

The fact that strata in these conditions can be correlated across faults, not constituting offlapping sediment that bypassed the slope topography, make them useful in the quan-

tification of cumulative tectonic uplift on the margin, an aspect addressed in Sect. 6 in this paper.

## 5.3 Vertically uplifted slope terraces and associated syn-rift topography

On the proximal parts of the western Iberian margin, correlative strata on the shelf and upper continental slope can be offset by several hundreds of metres by reverse faults and inversion structures (Figs. 4 and 5). This is most relevant when key stratigraphic markers are mapped across these inversion structures such as: (a) the Base Miocene unconformity recognized in seismic data by Mougenot et al. (1988) and by recent IODP Sites in SW Iberia; (b) the Top Cretaceous un-

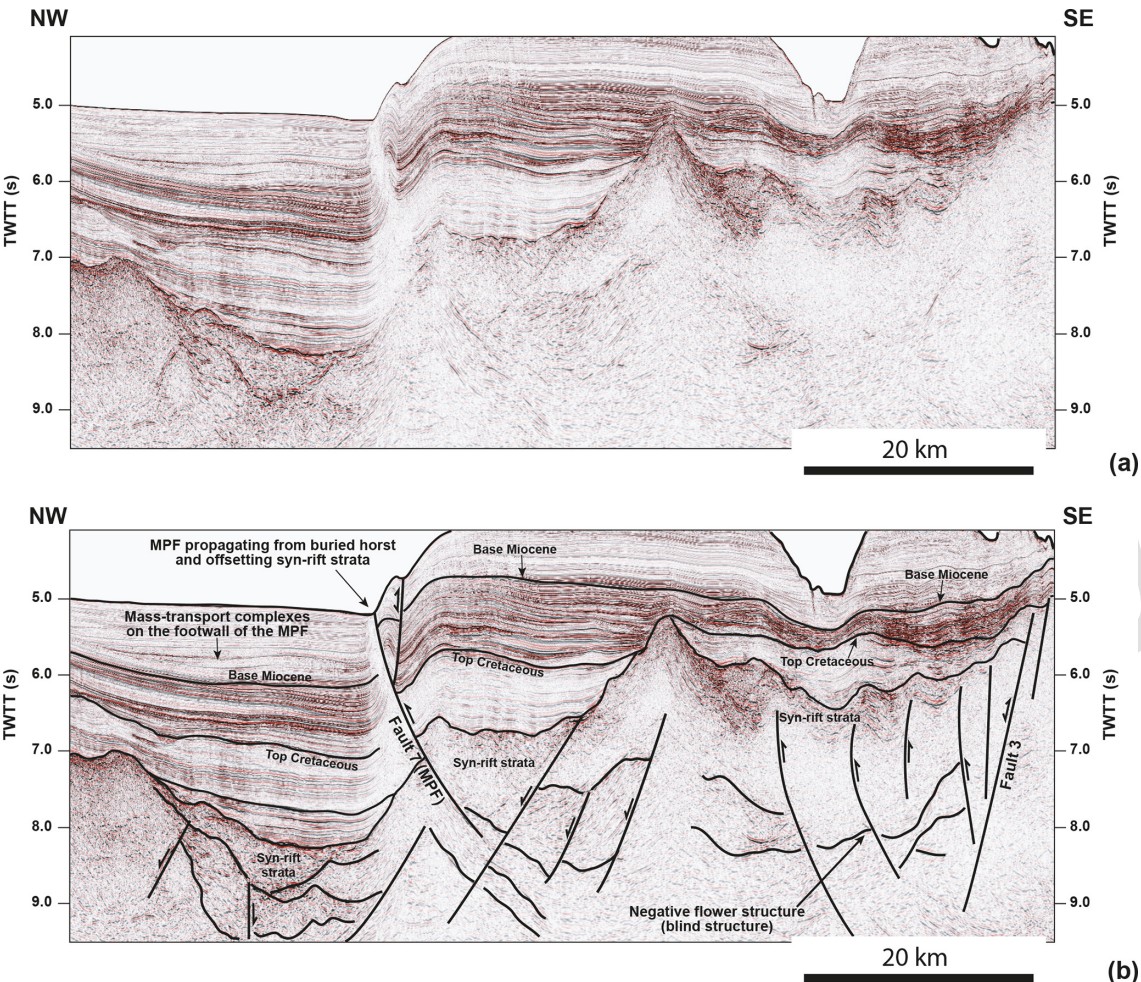

**Figure 5. (a)** Uninterpreted and **(b)** interpreted seismic profile across the Marquês de Pombal Fault (Fault 7) revealing this fault as offsetting syn-rift strata and rooting at a depth of ca. 9.0 two-way time (TWT). Vertical exaggeration reaches 6× on the profile shown. Note that not all syn-rift faults were reactivated during subsequent tectonic inversion. Also important is the presence of a negative flower structure in what is the southern tip of Fault 3. The location of the seismic profile is shown in Fig. 1. Seismic data are courtesy of TGS.

conformity, which is prominent all over SW Iberia; (c) the Cenomanian limestones of the Cacém Formation, which reveal a relatively constant thickness of 0.2 s two-way time (TWT), or 120–150 m, in the Lusitanian Basin and immediate continental shelf (see Alves et al., 2003, 2009); and (d) Upper Cretaceous sills and magma flows associated with buried magmatic complexes (Figs. 4, 5 and 8). At present, many of these structural terraces dip oceanwards from faults, revealing a mixed style of tectonic deformation that is akin to that described in Sect. 5.2.

Figures 4 and 8 show examples of uplifted terraces from distinct parts of western Iberia. Figure 4 reveals uplift of the shelf edge as correlative strata on the continental slope dip to the west due to tectonic shortening and uplift of the continental shelf as a whole. In Figs. 8 and B1 a similar geometry is observed in slope deposits, which respectively dip to the north and northwest and are deformed at depth. They occur

together with the marked folding and truncation of Cenozoic and Upper Cretaceous strata just below the seafloor. Other examples of deformed, uplifted syn-rift topography are mapped in several parts of SW Iberia as highlighted in the following section.

## 6  Quantification of tectonic uplift and horizontal advection

The tectonic framework of SW Iberia's basement is, at present, interrupted by magnetic anomalies, as shown in Fig. 6. These anomalies correlate with the presence of Late Cretaceous magmatic intrusions in the Central and SW sectors of western Iberia (Neres et al., 2023b). Complex magnetic anomalies are recognized near Lisbon and Sines, some of which comprise the largest magmatic bodies thus far identified in western Iberia (Neres et al., 2023b).

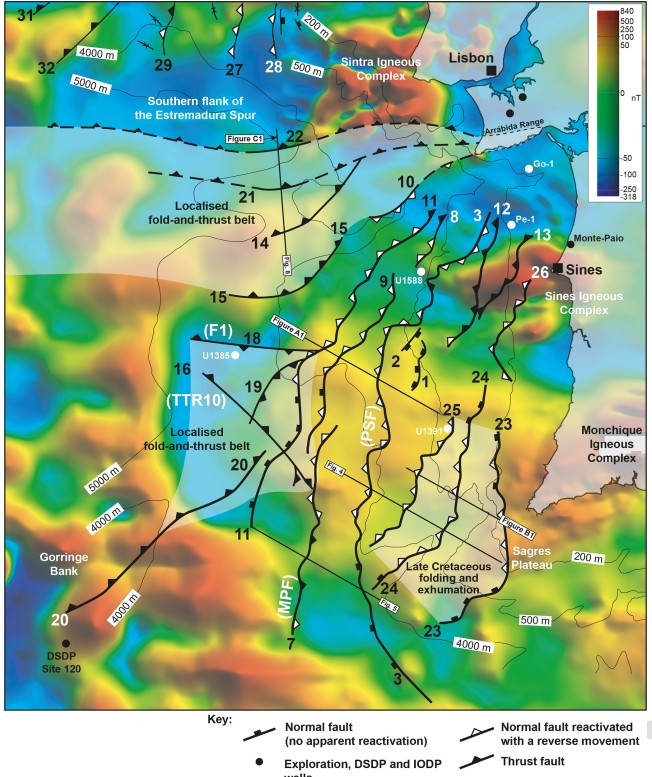

**Figure 6.** Structural map of SW Iberia superimposed on differential reduction to the pole (DRTP) data. The map, when combined with the interpreted seismic data, highlights the presence of localized, early stage fold-and-thrust belts between the Sintra and Sines magmatic complexes and west of this latter complex. Structures mapped are fault zones that are hard linked at depth to constitute > 200 km long features. Note that the white triangles indicate normal faults reactivated with a reverse movement, contrasting with the common black triangles, or "teeth", of thrust faults. DRTP data are provided by Getech UK.

Following the methodology in this paper, Fig. 9 shows a graphical representation of local uplift and horizontal advection associated with fault reactivation for the entire Atlantic margin of SW Iberia. The data plotted in Fig. 9 highlight important differences amongst the magnitude of uplift and horizontal advection recorded by the faults mapped in this work.

By comparing the maps in Fig. 9 with the graphs in Fig. 10 it becomes clear that the principal structures accommodating tectonic inversion in SW Iberia are Faults 3, 7 and the northern part of Fault 11. Of particular interest is the recognition of a corridor of deformation near the Sines Magmatic Complex and its offshore continuation (Figs. 6, 7 and 9). No major fault reactivation is recorded in the areas where these anomalies occur (e.g. Faults 12, 13, and 26). In contrast, all faults show enhanced uplift and horizontal advection west of Fault 12 and 23, and the offshore prolongation of the Sines and Monchique magmatic complexes (Fig. 9).

This is interpreted as proving a clear effect of sub-surface magmatic bodies on the magnitude of tectonic reactivation in SW Iberia, particularly west of the Sines Magmatic Complex and between this latter and the Estremadura Spur further north (Figs. 5, 8 and 9). In this more central region of western Iberia, the broad intrusion of Upper Cretaceous magma uplifted the so-called Estremadura Spur before the main phases of Cenozoic compression, and made this sector structurally higher (uplifted) in relation to the cooling, subsiding Tagus and Iberian Abyssal Plains that surround it.

Further south, the magnetic anomalies that extend offshore from Sines indicate the presence of a significant area intruded by Late Cretaceous magma, as recently recognized by Neres et al. (2023b) (Figs. 6 and 9). A wide number and variety of magmatic bodies were recognized by Neres et al. (2023b) and include kilometre-scale deeply intruded plutons to small plug-like and dike-like intrusions. The intrusion of these magmatic bodies was controlled by the crustal tectonic fabric inherited from the Paleozoic Variscan orogeny, which was later reactivated during Mesozoic rifting and subsequent Alpine collision. When interpreting the graphs and maps in Figs. 9 and 10, it becomes clear that the area intruded by this Upper Cretaceous magma records limited faulting and constitutes a structural buttress ahead of which most of the uplift and horizontal advection is accommodated by Fault 3 and major thrusts oceanwards from this latter structure.

Another key aspect is that the structures mapped in this work consist of large fault corridors at depth, essentially syn-rift normal fault strands that were hard-linked during Alpine-related compression. In contrast to previous data in Terrinha et al. (2003), the faults which accommodated most of the Cenozoic compression are not limited to frontal thrusts of a relatively shallow basal detachment. Instead, the set of faults occurring on the mid-continental slope – Faults 3, 7 and the northern part of Fault 11 – together record the greatest cumulative values of uplift and horizontal advection. Their lower tips are either rooted in (or offsetting) syn-rift strata or link to syn-rift faults at depths in excess of 9.0 s two-way travel time (∼ 10–12 km). In addition, Faults 23, 24 and 25 are also important structures accommodating strain to the northwest of the Monchique Magmatic Complex and the Sagres Plateau (Figs. 4 and 9).

## 7   Discussion

### 7.1   Structural controls on Alpine deformation offshore West Iberia

The data in this paper point to important tectonic reactivation offshore central and SW Iberia since the Late Cretaceous, first associated with the intrusion of magma in parts of its proximal margin and the Estremadura Spur and later via the reactivation of syn-rift faults, which became laterally linked structures (Fig. 8). Relatively large igneous edifices occur in

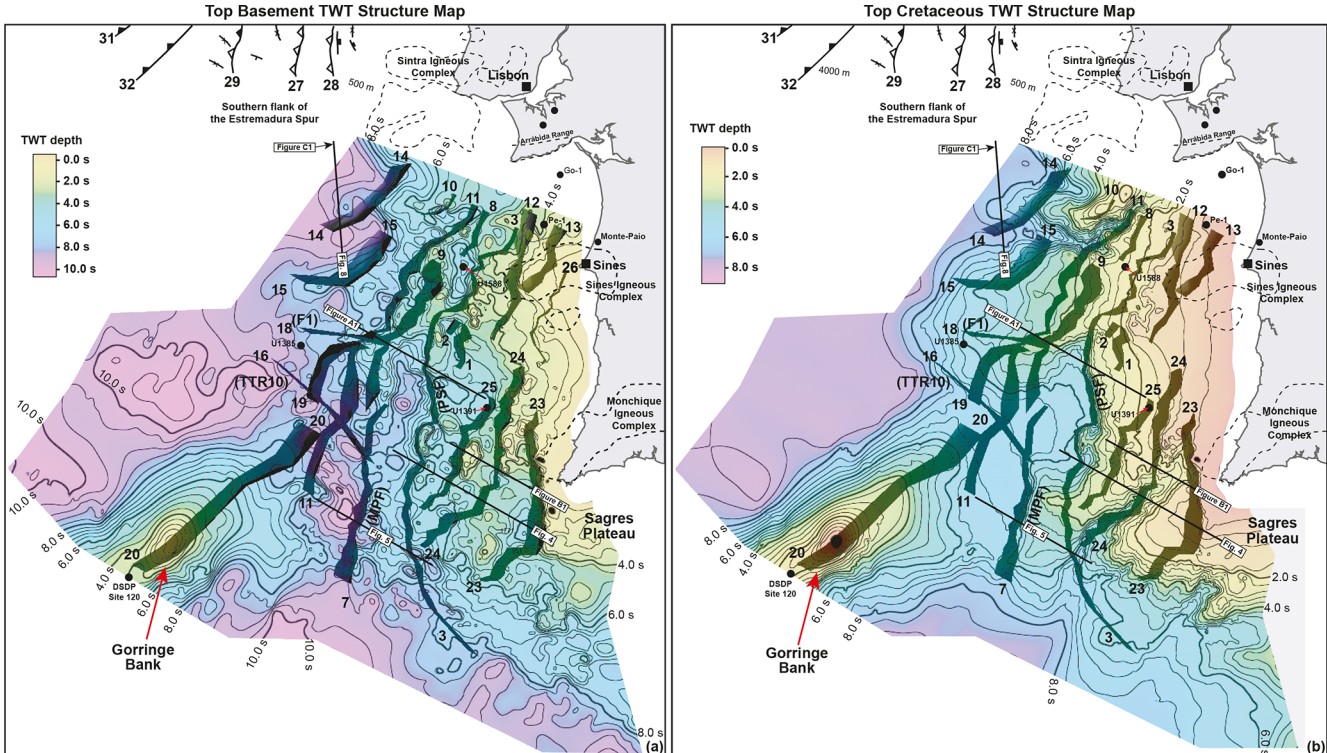

**Figure 7.** Two-way time (TWT) structure maps for specific intervals and reactivated faults interpreted in SW Iberia. **(a)** Top basement map indicating the presence of a shallower region of the margin to the east of Fault 3. **(b)** Top Cretaceous map highlighting that Fault 3 separates an area relatively deep to its west from a region to the east where exhumation and tectonic compression is of a higher magnitude on the SW Iberian margin. The contours of igneous complexes are also shown in the figure.

central Iberia, namely the Fontanelas Volcano (Pereira et al., 2021), together with other large buried magmatic bodies in SW Iberia (Fig. 6). The fact these magmatic bodies and associated sill complexes are known to be Late Cretaceous in age (Miranda et al., 2009; Pereira and Gamboa, 2023) makes them very good stratigraphic markers for quantifying uplift in the areas they are imaged.

This work interprets the effect of the largest of these igneous edifices to have been significant in the deformation history of SW Iberia. On the Estremadura Spur, a level difference of $\sim 4000\,\mathrm{m}$ is observed at present between the upper continental slope and the Tagus Abyssal Plain. The recognized pop-up structure that forms the Estremadura Spur at present may have been first thermally uplifted and then folded, accommodating a great part of this deformation on its southern and northern flanks (Fig. 7).

Deformation has also been accommodated at the base of the slope, south of the Estremadura Spur, by an early stage fold-and-thrust belt, as highlighted in Figs. 8, A3 and A4. The impact of magma intrusions in local strata deformation is also recorded at a local scale, with anticlinal structures and oversteepened, folded strata accompanying relatively deep intrusions (Pereira and Gamboa, 2023). A broader effect on local uplift and deformation is observed near where large

magma intrusions are located. East of Fault 3, towards Sines, a plateau region occurs with minor fault reactivation. Fault-related uplift and horizontal advection are much more pronounced west of this plateau, i.e. oceanwards from where Late Mesozoic igneous intrusions are recorded in seismic and magnetic data (see also Neres et al., 2023b) (Figs. 6 and 8). The effect of such promontory is discussed in more detail in the following section.

At a local scale, the several seismic profiles imaging reverse faults and thrusts demonstrate a close control of syn-rift structures on the growth and propagation of younger faults. In the particular case of Fig. 5, the Marquês the Pombal Fault is shown to be in part rooted on a syn-rift tilt block, with clear evidence for displacement of syn-rift units at the footwall tip of this same block. When this displacement is not clear in seismic data, strata near the Marquês de Pombal Fault reveal the propagation and development of splays of faults rooted on the tips of tilt blocks and other structural highs (Fig. 5). A similar structural style is observed offshore Lisbon, on the southern flank of the Estremadura Spur. Here, the putative NW–SE-oriented compression accommodated by base-of-slope strata is accommodated by a series of ENE–WSW thrust and reverse faults that root at the tips of syn-rift tilt blocks (Fig. 8). In addition, several sets of folds and thrusts

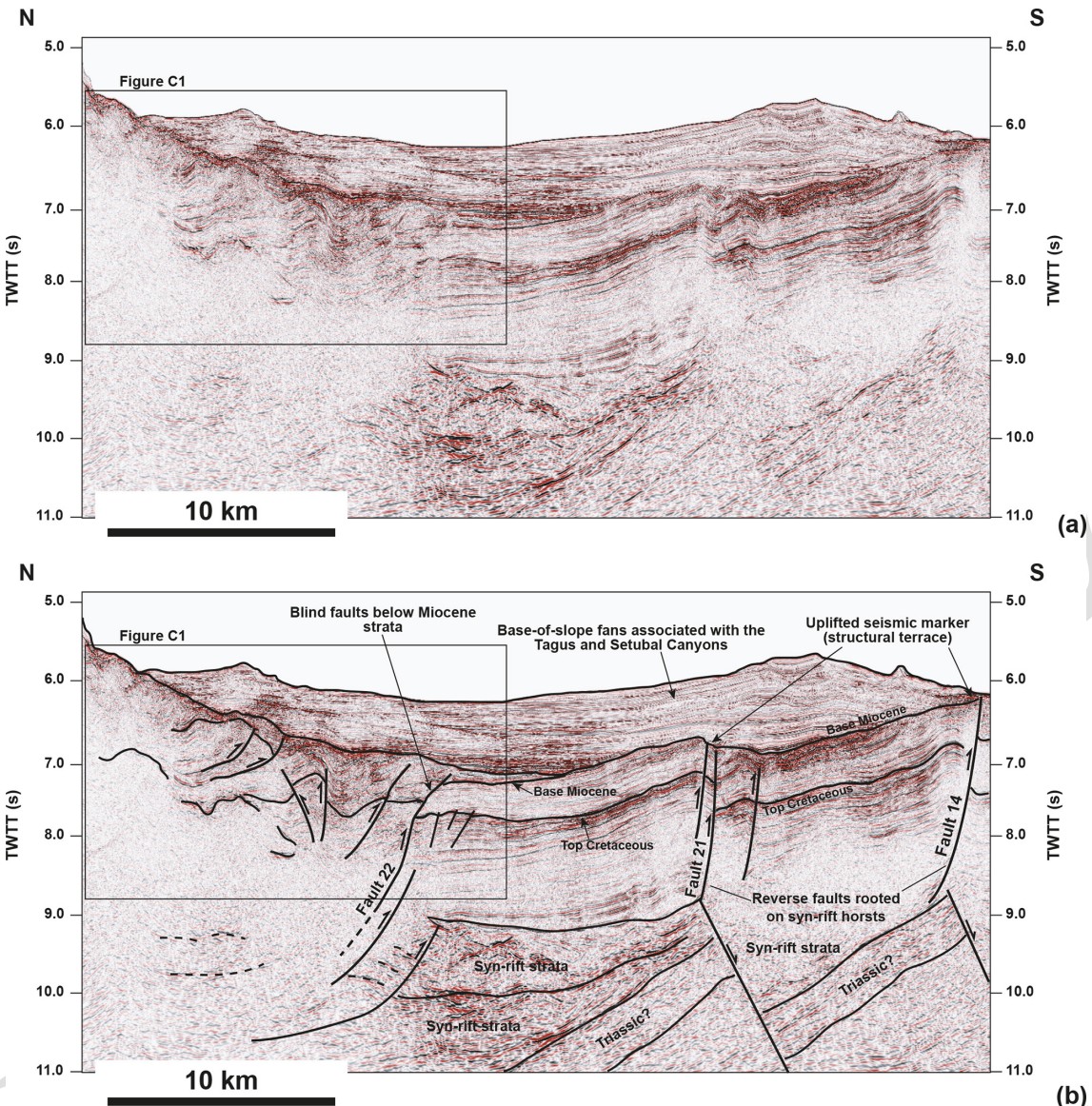

**Figure 8. (a)** Uninterpreted and **(b)** interpreted seismic profile from the southern flank of the Estremadura Spur highlighting the presence of a localized fold-and-thrust complex. This area of significant folding continues eastwards towards the Arrábida Chain, which is in the prolongation of Faults 14, 21 and 22 (see Fig. 6). Note the presence of blind faults below the Miocene–Holocene strata in the figure. Faults 14 and 21 propagate from the tip of syn-rift tilt blocks, which are observed at a depth of ca. 9.0 s two-way time (TWT). The profile is shown with a 6× vertical exaggeration and its location is in Fig. 1. Seismic data are courtesy of TGS.

are imaged in this same seismic profile in Fig. 8. Above the tilted syn-rift blocks, Mesozoic strata were deformed in a series of low-amplitude thrust anticlines and corresponding thrust faults, which are spaced at ∼ 20 km, replicating the spacing of syn-rift blocks below (Fig. 7). Above this first set of thrust anticlines occurs a more localized base-of-slope complex showing tight folding and deformation with a wavelength of ∼ 4 km (Figs. 8 and C1). Such an architecture resembles one of an early stage accretionary prism; in this case revealing a clear vergence of thrust anticlines to the south and a style of disharmonic folding that differs from

younger strata. While the deeper, lower-amplitude fold-and-thrust belt was developed as the offshore continuation of the onshore Arrábida Range – a folded succession of syn-rift deposits that structurally delimits the region south of Lisbon – the shallower complex is akin to gravitational complexes associated with transpressional tectonics that are recorded in Equatorial Brazil and southern Italy to cite two key examples (Davison et al., 2016; Mangano et al., 2023). This work thus postulates that such a complex folding results from distinct tectonic pulses associated with the Alpine orogeny. The older fold-and-thrust complex is capped by relatively under-

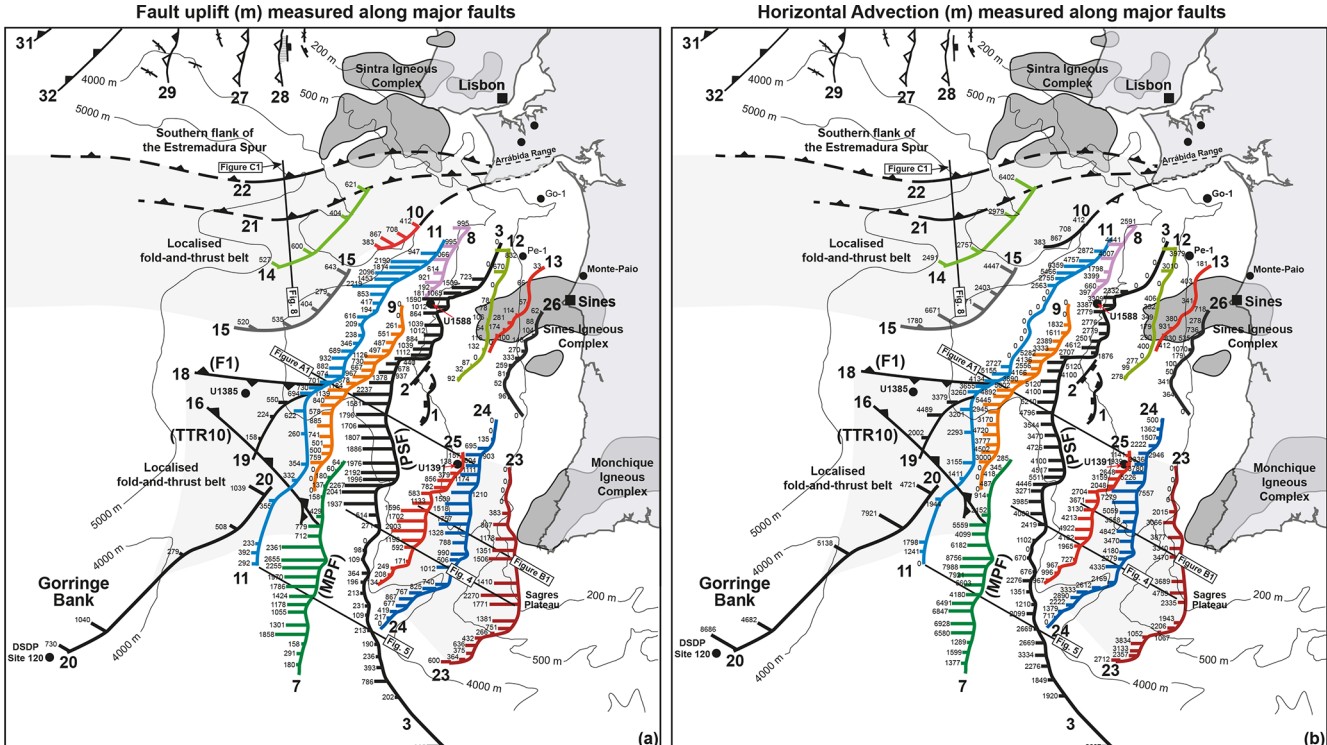

**Figure 9.** Graphical representation of fault uplift and horizontal advection as measured for each fault (see Supplement Table S1 and Fig. 10 for detailed data). The data show Faults 3, 7 and 11 as being the structures accommodating most of the Alpine compression in SW Iberia. Also important are Faults 24 and the joint Fault 14 and Fault 15 to the northwest of the study area.

formed Upper Cenozoic strata, which include base-of-slope deposits fed by the Cascais, Lisbon, and Setubal canyons, and shows Upper Cretaceous volcaniclastic sediment from the Lisbon Volcanic Complex deformed below a Cenozoic unconformity (Fig. 8). The smaller of the fold-and-thrust complexes is imaged above the latter volcaniclastic deposits, while Upper Cenozoic sediments are relatively undeformed (Figs. C1 and D1). It is therefore interpreted that the base of these underformed Cenozoic strata is Miocene in age (mid-Miocene?) and associated with the paroximal stage of Alpine tectonics in western Iberia (Cunha et al., 2019). The largest and most broadly spaced of reverse faults deformed latest Cretaceous and Paleogene strata and was first active during the Oligocene – either due to early stage tectonic deformation associated with the Betic compression phase or during the last episodes of Pyrenean tectonics. Importantly, these thrusts are seemingly active at present but were likely first formed during the earliest episodes of Cenozoic compression, perhaps starting during the terminal stages of Pyrenean tectonics, with main thrust and reverse faults having been reactivated in successive stages since then. Therefore, many (if not the most) of the inversion structures imaged in seismic data offshore SW Iberia are likely to have been reactivated in multiple episodes and potentially also as blind faults with no seafloor expression (Figs. 4, 5 and 8).

## 7.2 Significance of geometric coherence in adjacent reactivated faults

Geometric and kinematic coherence have been considered in the literature as proving the development of regular and systematic displacement patterns in related fault families (Walsh and Watterson, 1991; Walsh et al., 2003; Kim and Sanderson, 2005). Detailed measurements of uplift and horizontal advection for the faults mapped in SW Iberia reveal a clear geometric coherence (Figs. 9 and 10). Cumulative data for fault uplift and horizontal advection show typical coherent profiles in which the values recorded at the mid-part of SW Iberia, offshore Sines, are greater than on its northern and southern limits (Fig. 10a, b). The faults bordering the Estremadura Spur were excluded from our analysis as they show bathymetric differences of more than 4000 m, and part of this difference may be due to thermal cooling and subsidence of the Tagus Abyssal Plain relative to the Spur itself. In this particular case, it becomes hard to distinguish between the uplift component that results from tectonic- and magmatic-related uplift and that resulting from progressive thermal sinking of the two abyssal plains that limit the Estremadura Spur to the north and south.

In SW Iberia, Faults 3, 7 and the northern part of Fault 11 are confirmed as accommodating most of tectonic uplift, while Faults 7, 20 and 24 record the greatest horizontal ad-

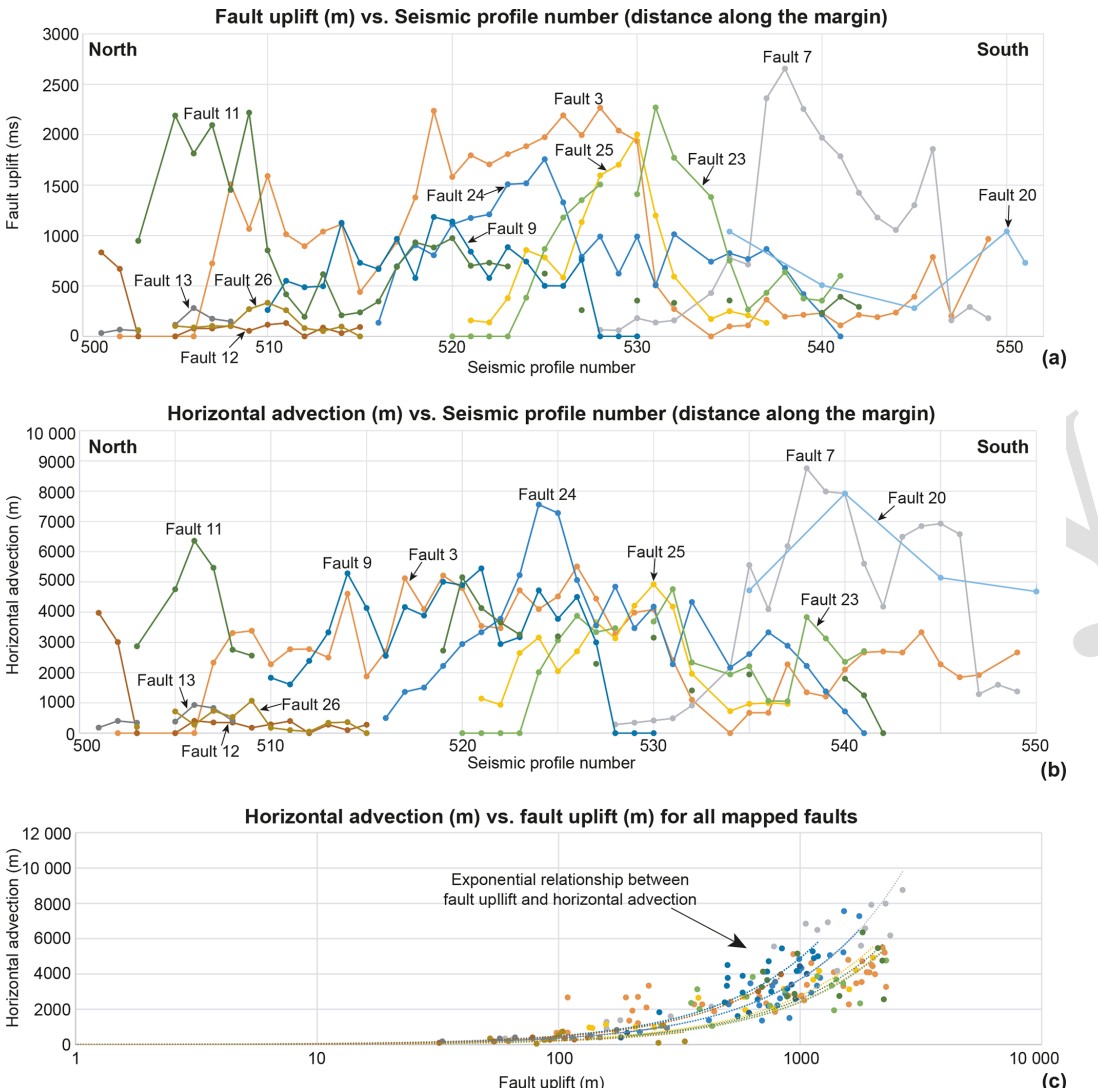

**Figure 10.** Graphs highlighting the distribution of fault uplift and horizontal advection in a north-to-south direction along SW Iberia, i.e. perpendicularly to the direction of tectonic shortening and along the faults mapped in this work. **(a)** Relatively large values of uplift are recorded for Faults 3, 7 and 11. **(b)** Horizontal advection is larger for Faults 7, 11 and 24, with Fault 20 also presenting a significant value. **(c)** Plot of horizontal advection (m) vs. fault uplift (m) for all mapped faults. A $V_p$ velocity of $2000\,\mathrm{m\,s^{-1}}$ was used to convert uplift magnitudes in milliseconds (ms) to their corresponding values in metres (m). The graph highlights the exponential relationships between these two properties for all the structures mapped.

vection (Figs. 9 and 10). In the particular case of Fault 3, the Slope Fault System of Alves et al. (2009), Fig. 9 reveals it as a coherent structure along its strike that is kinematically linked through a distance of $\sim 200\,\mathrm{km}$. Such a characteris-
5 tic, together with the similar profiles for cumulative uplift and horizontal advection, are clear indicators of geometric and kinematic coherence. The cumulative data in Fig. 11a and b also show a typical C-shaped profile along SW Iberia, which is typical of coherent fault networks. This naturally
10 means that not all the faults in this network will necessarily be reactivated in a single seismic event, but that the rate in which the all systems of faults grew is coherent and reflects

the development of a linked fault network in SW Iberia. It also suggests that the sequential movement of this fault network is an important phenomenon in the study area, con- 15
firming the postulate of Walsh and Watterson (1991). They stated that forward, rearward and lateral propagation in fault arrays are equally important when fault coherence is confirmed, in many ways replicating the setting of convergent margins such as SE and E Japan (Tsuji et al., 2014; Kimura 20
et al., 2018) and other areas recording significant tectonic shortening. In parallel, the quantification of cumulative uplift and horizontal advection values in this work also indicates that the magnitudes of tectonic reactivation were greater near

structural barriers (buttresses) between the Estremadura Spur and Sines than to the northwest of the offshore prolongation of the Monchique Magmatic Complex. There is a difference in cumulative uplift and horizontal advection as one reaches these regions.

Comparing Figs. 6 and 9 with the fault data in Fig. 10 provides a robust correlation amongst the areas affected by important tectonic compression and the Sines Magmatic Complex. Note that several igneous edifices occur west of the Sines Magmatic Complex in SW Iberia. These igneous edifices are also imaged in seismic data, coincide with local horsts and seamounts (e.g. Descobridores Seamounts), and are often bounded by thrust and reverse faults propagating from deeper parts of the crust to delimit a proximal sector of SW Iberia that was tectonically uplifted in the Late Cretaceous-Cenozoic.

### 7.3 Implications for future geohazard assessment

A first major result of this work is that a coherent fault mode hints at the possibility of reactivating SW Iberia through a wide area during large seismic events. In such a setting, previous fault configurations, limited to proposing the lateral reactivation of two-three faults strands, are further complemented in this work by evoking the possibility of reactivating faults, and associated structures, that are sequentially placed forward or rearwards of the MPF and TTR10 faults mapped by Terrinha et al. (2003). Thus, the model in this work suggests that fault displacement can be kinematically accommodated by multiple structures during a large seismic event, increasing the seismogenic and tsunamigenic potentials of SW Iberia. The importance of this observation lies on the seismic magnitude–fault length relationships recognized for active faults; when fault length is properly constrained by geophysical and geological methods, it is broadly understood that the potential maximum earthquake magnitude correlates positively with rupture length and generically with fault length (Bohnhoff et al., 2016; Trippetta et al., 2019). The recognition of a > 200 km long Fault 3, for instance, suggest the potential to generate $M_\mathrm{w}$ 8.0 earthquakes in this structure alone (Wyss, 1979; Bonilla et al., 1984; Trippetta et al., 2019).

The presence of structural buttresses landwards from the continental slope, at the approximate latitude of Sines and near the Sagres Plateau, also has implications regarding the seismogenic potential of SW Iberia. This work suggests that areas where Late Cretaceous magmatism was more intense and magma intruded the crust in greater volumes are structurally harder and more stable than the regions not affected by this magmatism. They were also thermally and mechanically uplifted at the start of the Alpine orogeny, generating structural "indentors" – namely parts of the continental margin that formed hard buttresses and were relatively underformed by the tectonic shortening imposed on SW Iberia, in a N–S and NW–SE direction, during the Cenozoic. This paper also highlights the presence of immature fold-and-thrust belts south of the Estremadura Spur, itself interpreted as a large pop-up structure by Ribeiro et al. (1990), and west of the Sines and Monchique magmatic complexes (Figs. 6, 7 and 11c, d). By the same token, the presence of a hard magmatic "core" of rock offshore Sines and Monchique, likely associated with the thickening of crust at its base, may provide the reasons for why the faults located to the west of the SW Iberia indentor accumulated the bulk of uplift and horizontal advection. Finally, it may also explain why Alpine tectonics seems to be in its early stages of forming a subduction zone in SW Iberia, with this hard indentor concentrating strain under a prolonged Late Cenozoic setting dominated by NW–SE tectonic compression (Ribeiro et al., 1996; Neres et al., 2016, 2018; Somoza et al., 2021). As a corollary, it is recognized here that the presence of this indentor led to a cumulative tectonic uplift of > 6 km in the middle part of SW Iberia's Atlantic margin since the Late Cretaceous (see cumulative uplift values in Fig. 11a).

## 8 Conclusions

Seismic and borehole data were used to quantify, for the first time, the true magnitude of tectonic uplift and inversion experienced by the Atlantic margin of SW Iberia. The recognition of geometric coherence in the faults mapped in SW Iberia hints at a degree of synchronous movement thus far not proven in the literature. The main conclusions of this study can be summarized as follows.

1. The recognition of structurally offset and oversteepened stratigraphic markers in SW Iberia demonstrates a magnitude of uplift of the Iberian Plate core that is, cumulatively, greater than previously assumed. Often assumed to be on the order of 1–1.5 km, the amount of Cenozoic tectonic uplift recorded by some of the faults mapped in SW Iberia exceeds 2 km and cumulatively can reflect a total uplift of > 6 km since the Late Cretaceous.

2. Magmatic edifices and the intrusion of magma below the Late Cretaceous crust and nearby basins led to the thermal uplift of large areas of SW Iberia. These areas were later reutilized as structural buttresses to Alpine compression, with the most developed inversion structures being developed around these buttresses.

3. Geometric coherence in faults reveal these can be reactivated in tandem during the largest of seismic events, thus enhancing the seismogenic and tsunamigenic potential of SW Iberia. Faults reveal geometric coherence along the margin and, putatively, may be kinematically coherent when of the largest seismic events. Importantly, Fault 3 comprises a ∼ 200 km long fault zone with the potential of generating $M_\mathrm{w}$ 8.0 earthquakes.

4. The presence of a hard magmatic "core" of rock in the middle part of SW Iberia, near Sines, justifies the for-

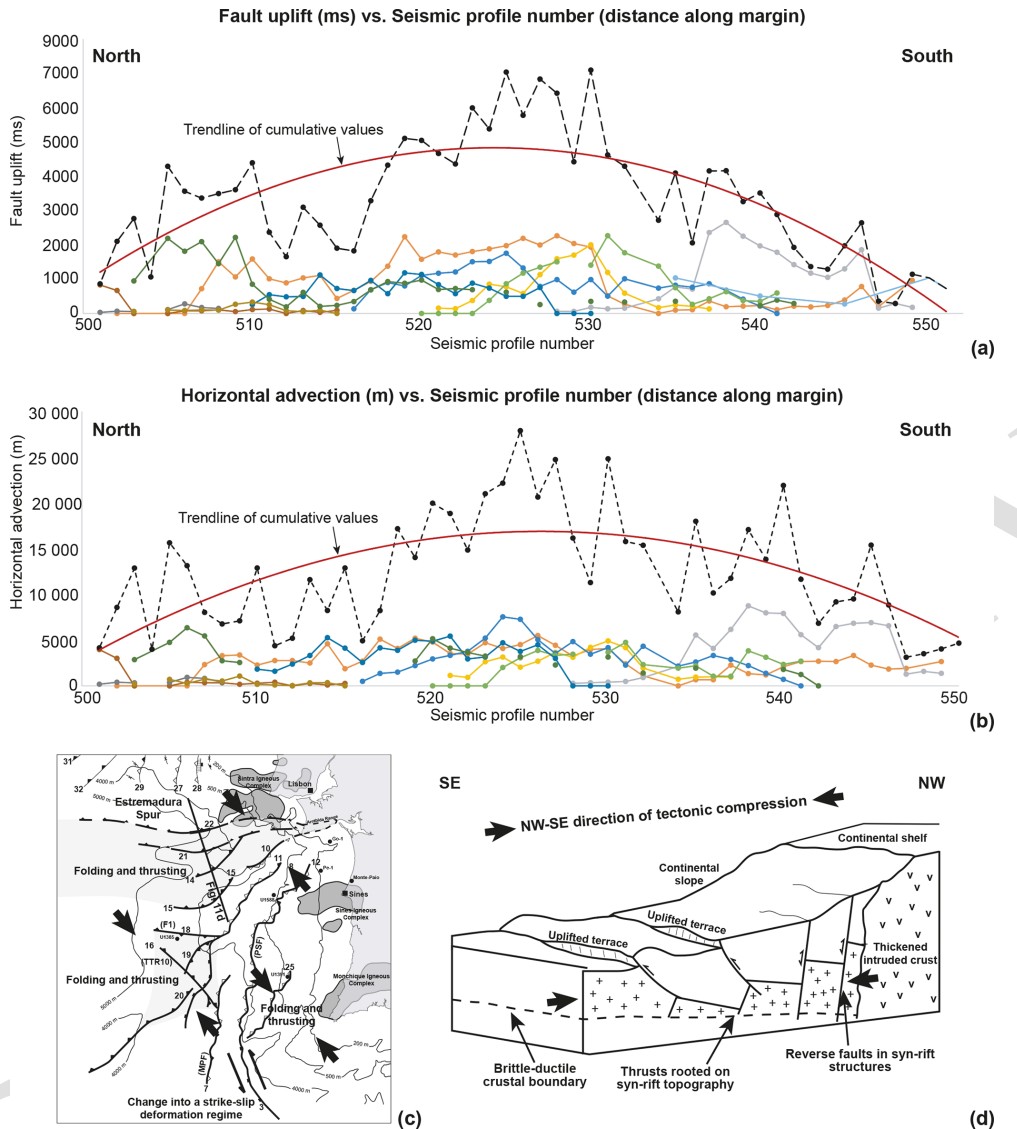

**Figure 11.** Cumulative data for fault uplift and horizontal advection as measured from north to south along SW Iberia, i.e. perpendicularly to the direction of tectonic shortening and along the faults mapped in this work. **(a)** A typical C-shaped curve for uplift is observed when plotted against distance. **(b)** A similar C-shaped curve is observed for horizontal advection, mimicking the results in the first graph (Fig. 11a). **(c)** Regional map summarizing the tectonic setting observed in SW Iberia and justifying the generation of localized fold-and-thrust belts. **(d)** A 3D block diagram summarizing the reactivation style of faults in the study area against a hard crustal buttress formed around the areas intruded by Late Cretaceous magma.

mation of a structural indentor and why the faults located to the west accumulated the bulk of uplift and horizontal advection. It also explains why Alpine tectonics is slowly progressing to form a fully developed subduction zone in the study area, with this hard indentor focusing tectonic deformation ahead of a thickened, hard intruded part of the western Iberian margin. Similar uplifted, structural buttresses coincide with the Estremadura Spur in central Portugal and the Sagres Plateau in the Algarve.

**Appendix A**

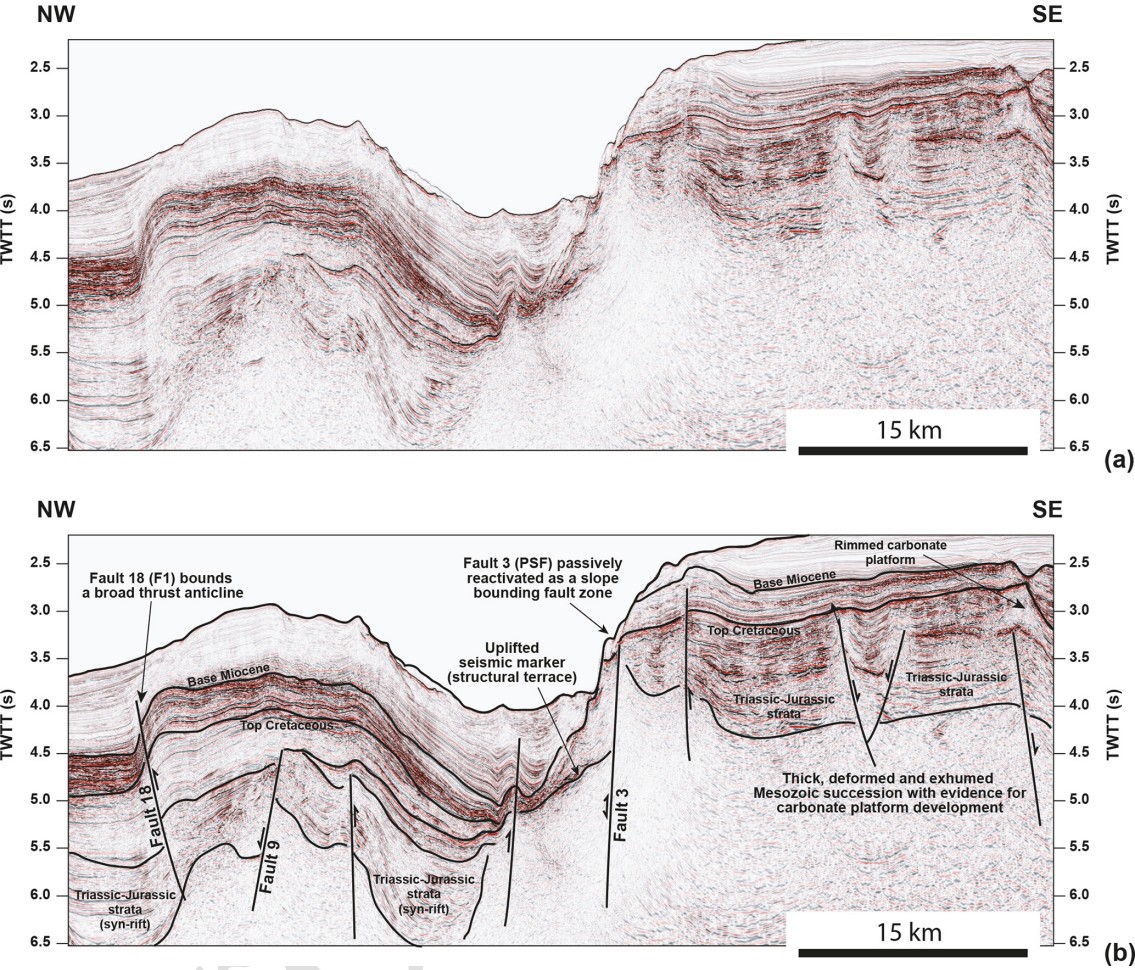

**Figure A1. (a)** Uninterpreted and **(b)** interpreted seismic profile from SW Iberia highlighting a vast area of Late Cretaceous erosion and exhumation to the east (i.e. landwards) from Fault 3. The profile is shown with a $6\times$ vertical exaggeration. The region oceanwards from this same fault reveals a gentle thrust anticline and evidence for shortening at upper crustal level. Note that Faults 3 and 18 are rooted in basement rocks and are likely related to the reactivation of deep crustal structures inherited from the syn-rift stage. The location of the seismic profile is shown in Fig. 1. Seismic data are courtesy of TGS.

**Appendix B**

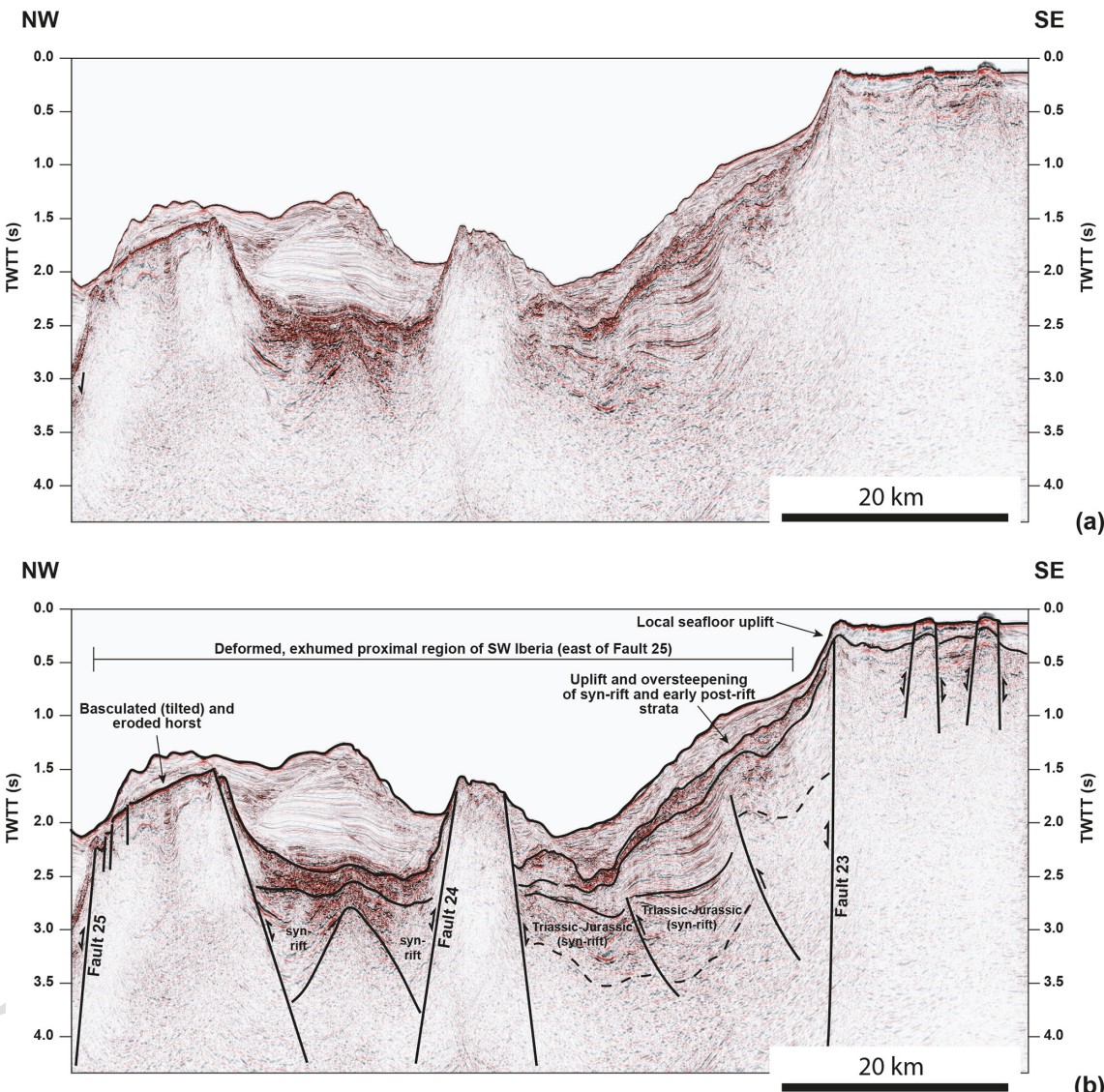

**Figure B1. (a)** Uninterpreted and **(b)** interpreted seismic profile from SW Iberia imaging the area next to Faults 23, 24 and 25, where important uplift is recorded in Mesozoic strata. The profile is shown with a 6× vertical exaggeration. These strata are oversteepened and uplifted near the most active faults in the study area, e.g. Fault 23 and the whole region to the east of Fault 25. The location of the seismic profile is shown in Fig. 1. Seismic data are courtesy of TGS.

**Appendix C**

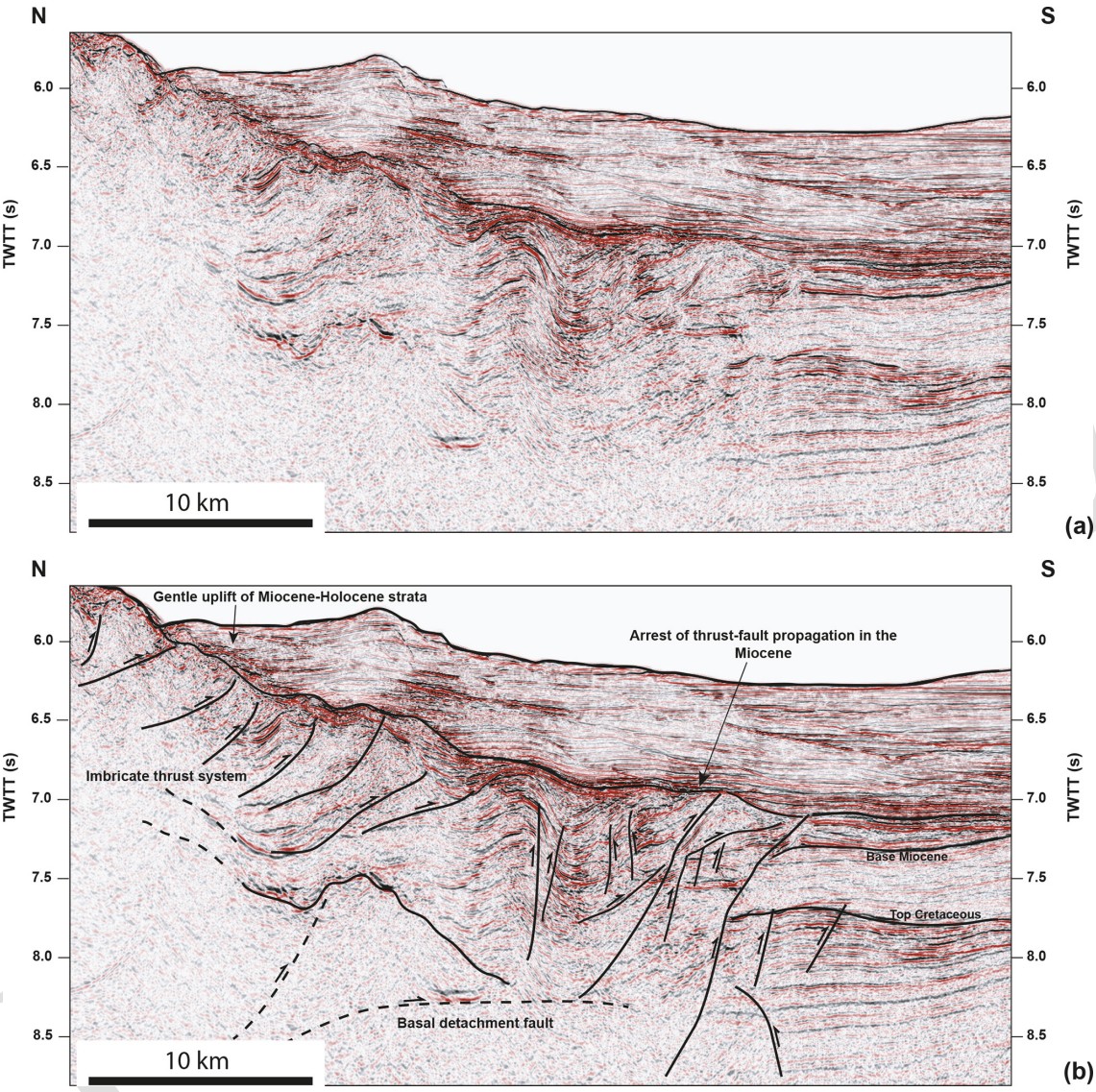

**Figure C1. (a)** Uninterpreted and **(b)** interpreted seismic imaging the southern flank of the Estremadura Spur and its youngest fold-and-thrust complex. The profile is shown with a 6× vertical exaggeration. Note the arrest of the folding and thrusting during the Miocene, which indirectly dates of the sediment apron west of Lisbon as Miocene to Holocene in age. The location of the seismic profile is shown in Fig. 1. Seismic data are courtesy of TGS.

## Appendix D

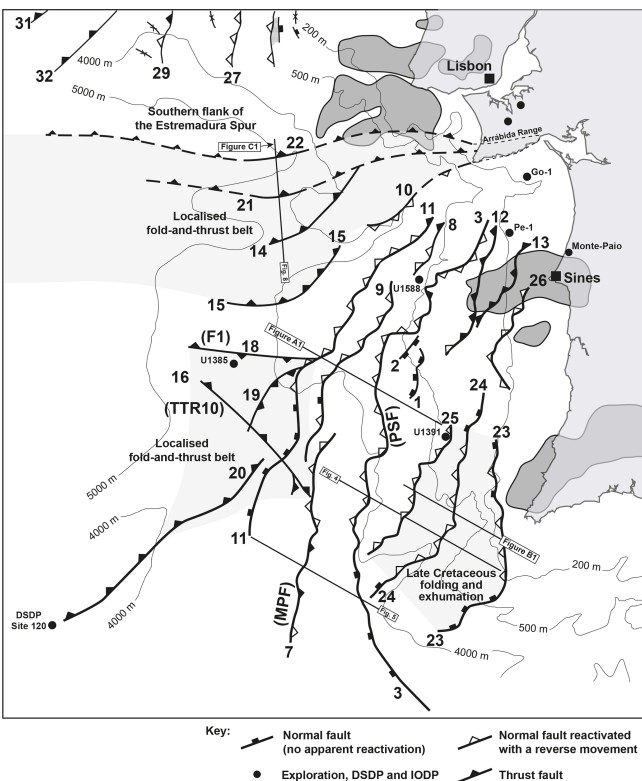

**Figure D1.** Structural map of SW Iberia highlighting the main structures and their reactivation histories. Structures mapped constitute long fault zones that are hard linked at depth to constitute > 200 km long features, as in the case of Fault 3.

*Data availability.* The seismic data in this paper were provided by TGS and are available upon request. Getech provided the magnetic data in Fig. 6, which are also available upon request. Bathymetric and seismological data for western Iberia, i.e. the locations and relative magnitudes of earthquakes for the period spanning 1900 to 2006, were obtained from GeoMapApp (http://www.geomapapp.org, Ryan et al., 2009).

*Supplement.* The supplement related to this article is available online at: https://doi.org/10.5194/se-15-1-2024-supplement.

*Competing interests.* The author has declared that there are no competing interests.

ther geographical representation in this paper. While Copernicus Publications makes ev-

ery effort to include appropriate place names, the final responsibility lies with the authors.

*Acknowledgements.* The author would like to thank the reviewers Gang Rao and Óscar Fernández for their constructive comments. Marta Neres is acknowledged for her constructive "offline" comments and advice on the most recent geophysical data acquired on the SW Iberian margin. Schlumberger supported this study and the 3D Seismic Lab at Cardiff University. TGS is acknowledged for the provision of new, reprocessed seismic data from western Iberia.

*Review statement.* This paper was edited by Yang Chu and reviewed by Gang Rao and Oscar Fernandez.

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
