# Peer review of "Networks of geometrically coherent faults accommodate Alpine tectonic inversion offshore SW Iberia"

_EGUsphere, 2023_

## Referee Comment (RC2)

Dear editor,

In the article "Networks of geometrically coherent faults accommodate Alpine tectonic inversion offshore SW Iberia", the author presents a beautiful and excellent quality set of reflection seismic images of the Alentejo offshore basin, in SW Iberia. He proceeds to use his interpretation of the seismic profiles to propose an idea of coherent and systematic inversion of a fault network in a passive margin and the influence of crustal heterogeneity on this pattern. This a valuable idea that could be tested on other margins (including some nearby ones!) with potentially very significant implications. He further raises the prospect of the analysis highlighting potential seismic risks for this region, which is a highly valuable contribution to risk mitigation. The article is very well written, and is very easy to follow. I do however, have some significant issues with the interpretation of the seismic profiles presented, as the fact that they are in the time domain has been mostly ignored (or so it seems to me) and the interpretation could be driven significantly differently if these lines were interpreted in depth. In that sense, I think the author needs to make a much better case to support/argue for his interpretation and to discuss what uncertainties may underlie it.

best regards,

Oscar Fernandez

**Major issues:**

Vertical exaggeration is major on the seismic profiles. It is misleading not to include even an estimate of what this could be. On Fig. 5, for instance, assuming a high velocity of 4000 m/s for the whole line, we would have a vertical exaggeration of 3.3x (Fig.4 and Fig. 7 have around 1.7-1.8x exaggeration with that same velocity; App. A1 is 3.1x exaggerated; App. A2 is a whooping 4.7x exaggerated if not more!). Naturally, vertical exaggeration depends on the velocity and since shallow velocities can be very low, then the shallower portions (water velocity 1500m/s) are highly exaggerated and the deeper ones are possibly not as exaggerated as calculated above (basement velocity can be 5-6000 m/s or higher). Petrel and other software tools provide excellent options for depth conversion. I strongly encourage the author to provide even a first-pass depth conversion (water at 1500 m/s would make a HUGE difference in how the uplift-related bathymetry is perceived) and in its defect an estimate of vertical exaggeration. Just for the sake of demonstrating the importance of not looking at refelction seismic only in the time domain, an illustration of the lines in Fig 5 and Appendix A1 are shown below, scaled very approximately so that every second TWTT is equal to 2 km (4km/s seismic velocity). I think the difference in how one could interpret these lines is clear. Of course, the author has traced these structures across multiple lines for consistency, and has even uncovered internal coherence in his interpretation, but I find a much better case needs to be done to convince the reader that the interpretation is actually correct. Sorry Tiago for being a pain with this: I think we really need to be extremely careful with seismic! No ill intention meant, but I would really expect to see a better clarification of why you have picked faults the way you have, and what uncertainties there may be in the interpretation and what impact (if any, of course) this would have on the rest of the analysis.

The same applies to the analysis of throw and heave (Section 6). You are combining different magnitudes (time and true distance). It is well worth clarifying this and stating any possible caveats.

[Figure]

On the line in Fig. 7, the depth in time goes down to 11s TWTT. This depth is huge! Most likely, some of the reflectors you are seeing could even be Moho!!! I know the Estremadura spur is likely not in isostatic balance (held up by far field stresses, which only makes things worse, actually), but in isostatic balance, your Moho reflections tend to come in at depths of 9-11s TWTT!! See the paper of Warner (1987, https://doi.org/10.1111/j.1365-246X.1987.tb06651.x). I encourage you to have a look at this line again and see if there is something even more spectacular that you may have missed!!! :)

Fault 3: I understand this becomes a key structure in your analysis. I have, however, a question as to whether this is really a single fault, and as to whether it really is an extensional fault. On Fig. 4 you clearly identify it as a SE dipping thrust. Why do you not interpret it the same way on Appendix A1? I think it would make much more sense! In any case, if you interpret the fault to change from extensional (transtensional?) to thrust along strike, then you shouldn't really interpret it as being linked. If you observe the throw map for this fault (Fig. 8a, 9a) it seems clear to me that it could actually be formed by three independent segments. As you are working on 2D seismic, I would like to see a better argumentation in favor of this being a single, long (linked-up?) fault.

I am sorely missing a structural map (i.e. isobath or isochron map) of a pre-uplift horizon. The map in Fig. 6 is misleading because the fault cutoffs shown are not referenced to any specific interpretation horizon. The Base of Miocene is a likely candidate. The reason that I ask this is because the structural map is a very quick way of visually assessing the relative throw on each fault. I think this would be a major positive contribution to the article – this helps remove ambiguity on your fault interpretation in a very significant way.

There is a major argumentation flaw (in my opinion) in applying some of the criteria in section 3.2 further down. In Section 5.2 you talk about folded strata and you previously mentioned in section 3.2 that stratal thickness is an important criterion in distinguishing post-rift from syn-rift fault activity. However, your seismic images are shown in time, with partially extreme vertical exaggeration (the values I calculated above are low range values for your sediments, that likely have seismic velocities lower than 4km/s and are therefore more vertically exaggerated). You can only use this criterion if you somehow attempt to display these same structures in depth. In my view, as this is, it is not admissible (sorry for the harshness, no ill intention).

Salt tectonics is prevalent both to the north and to the south (Pena dos Reis et al. 2017, https://doi.org/10.1016/B978-0-12-809417-4.00015-X; Cascao et al. 2023, http://dx.doi.org/10.1306/08072221100; Matias et al. 2011, http://dx.doi.org/10.1306/01271110032; Ramos et al. 2017a, https://doi.org/10.1016/j.marpetgeo.2017.09.028) but is not interpreted here. I understand it has

been previously stated that this basin is peculiar because of this absence of salt tectonism. However, on the seismic data provided, there are isolated features that may be diapirs based on their post-rift and pre-uplift growth strata and short wavelength. Most evident may be the structure under the text "blind thrust" above Fault 25 on Fig. 4b. It is somewhat reminiscent of the wavelength of structures in the westernmost Algarve basin (cf. Fig. 12 in Ramos et al. 2017a). Furthermore, even if the Dagorda salt was not a player in syn-rift salt tectonics, it would most likely have acted as a detachment during shortening. Indeed, your "thrust-and-fold-belt geometries" are somewhat reminiscent of fold belts with a well expressed detachment.

In terms of terminology, I understand the use uplift and advection for rock masses, but when you are talking of displacement on faults, the terms that are normally used are throw and heave. Is there a reason not to use these? Furthermore, when you give values for throw and heave, you should refer them to the specific marker/horizon/stratigraphic interval for which you have measured the offset.

**Minor comments:**

line 50 – I understand it lies "around the corner" but a comparison to the well documented example of Ramos et al. 2017b (https://doi.org/10.1002/2016TC004262) could be interesting

line 91 – "in what ways do tectonic"

line 92 – "and how do they interact"

line 101 – please check the recent publication of Amigo Marx et al. 2022 (https://doi.org/10.1111/ter.12595)

line 146 – there is an event of re-activation of salt structures in the Algarve basin in the Early Cretaceous and related sedimentary record (Ramos et al. 2016, https://doi.org/10.1007/s00531-015-1280-1) that likely relates to the regional picture (note that

line 155 – inversion in the Algarve extends beyond the Miocene (Ramos et al. 2016, 2017b)

line 183 – He et al. missing year

line 187-193 – slight repetition

line 197-198 – beds are not really deposited horizontally on continental margins (the syn-uplift units on your own seismic betray you) ;) I think it is worth stating it is near-horizontal (my personal battle against assuming pre-deformation = horizontal)

line 217 – the faults on Fig. 6 look like they are NE-SW striking, not NW-SE. Can you please check this is correct? Also, if your faults indeed strike NW-SE then sections in Fig. 4 and Fig. 5 would be parallel to your fault, which I think is not what you interpret?

line 219 – "on seismic data"

line 228-29 – NW-dipping normal faults reactivated as SE-dipping reverse faults is a pretty paradoxical statement... I understand it, but I think it would be useful to discuss this style of inversion (compare to the very similar style of inversion described by Ramos et al. 2017b for the Guadalquivir and Portimao Banks in Algarve). As in the example of Algarve, I think this speaks for an inherited basement fabric and could speak for both basins sitting on a similar SPZ-style basement (with inherited Variscan thrust fabric that re-activates as opposed to inversion of normal faults). (?)

line 267 – to speak of 100s of meters, you should provide a vertical scale in meters.

Fig. 6 – can you please check Appendix B and Fig. 4 are correctly placed here? There is a line labelled SF2 which seems to be App B, and App B seems to be Fig. 4? If Fig. 4 is not on this map, please locate it.

Fig. 6 – Fault 3 on Fig. 4 is a SE-dipping thrust, but on Fig. 6 it is represented as a NW-dipping thrust/inverted fault.

Section 6 – the analysis here is mixing depth in TWT milliseconds with horizontal distance in meters. This requires some very careful clarification.

line 320 – provide a Fig. reference for faults 3,7,11. (Fig. 5?)

line 329 – sorry, not sure where you see that the intrusions drive inversion? Did you really explain it before? Wasn't clear to me at least. OK, is it what you later say in line 340? Then you need a reference here – it is not something you are demonstrating with your data.

line 340 and further – if igneous bodies are so important in your inversion history, it is worthwhile including a seismic line that shows these relationships

lines 357-358 – deformation is really mostly accommodated by the folds on the left part of Fig. 7. The thrusts that you interpret linking to the extensional faults are really minor. They don't really accommodate much shortening... furthermore, I am not fully convinced that what you are seeing at the base of the thrusts is really syn-rift strata and not some form of deep crustal reflectors.

line 390 – I don't fully follow this argument. Thermal subsidence is well known for passive margins. I do not know that it applies to inversion setting (!?) nor to intrusions (whose thermal-related dilation potential is probable limited).

Fig. 8 – you place your throw and heave bars always on the western side of the faults. However, I find this misleading and I think it would be much more accurate to place them always in the hangingwall or always in the footwall, or any other placement that makes kinematic sense (block up vs block down). But placing them always west seems to indicate these are the upthrown or downthrown blocks consistently, which is not the case.

Fig. 9 – you should NOT plot depth vs time domain units. This is not valid. Please revise, provide an approximate conversion  factor or something similar before you attempt this.

Fig. 9, 10 – it is not clear what your horizontal axis is. It needs to be clarified and made evident in a location map. I assume you are referring to dip seismic profiles? Are they parallel to your assumed shortening direction? If not, then you cannot really add up displacements... You need to state these assumptions better.

line 430 – this is an interesting thought. However, shortening should be accommodated along strike equally. What happens to the shortening that is not expressed around the intrusions? Is this internal strain or is it being transferred elsewhere?

lines 459-463 – this is very interesting point and perhaps is worth bringing into the discussion. After all, this provides a mechanism to accommodate the type of dynamic topography implied by the lithospheric folding model (Cloetingh et al. 2002, https://doi.org/10.1029/2001TC901031)

line 465 – I do not see that the authors have documented intrusion-related uplift... Sounds like a conjecture? Please check this is adequately argumented.

---

## Author Comment (AC2)

Dear Editor and Reviewer #2 (Oscar Fernandez)

I am happy to present, with this letter, a detailed reply to the constructive comments provided by Reviewer #2 on both the interpretation and content of parts of the paper. I first need to thank the reviewer for the thorough analysis of my paper, and for questioning crucial points regarding both its narrative and figures. The comments provided are pertinent and coincide with questions I had posed myself when first starting this work, when making sense of the new re-processed seismic data from the SW Iberian margin.

I provide below a point-by-point reply to Oscar Fernandez' queries.

**Key comments**

**1) In the article "Networks of geometrically coherent faults accommodate Alpine tectonic inversion offshore SW Iberia", the author presents a beautiful and excellent quality set of reflection seismic images of the Alentejo offshore basin, in SW Iberia. He proceeds to use his interpretation of the seismic profiles to propose an idea of coherent and systematic inversion of a fault network in a passive margin and the influence of crustal heterogeneity on this pattern. This a valuable idea that could be tested on other margins (including some nearby ones!) with potentially very significant implications. He further raises the prospect of the analysis highlighting potential seismic risks for this region, which is a highly valuable contribution to risk mitigation. The article is very well written, and is very easy to follow.**

**Reply:** Thank you for the positive comments below. By the time of this review, I had already completed a review of grammar and typos after receiving first comments from Reviewer #1. I believe the new comments now provided by Reviewer #2 will re-enforce this manuscript's quality – there were some typos left to be addressed after the first review.

**2) I do however, have some significant issues with the interpretation of the seismic profiles presented, as the fact that they are in the time domain has been mostly ignored (or so it seems to me) and the interpretation could be driven significantly differently if these lines were interpreted in depth. In that sense, I think the author needs to make a much better case to support/argue for his interpretation and to discuss what uncertainties may underlie it.**

**Reply:** The fact the data were displayed in time was not ignored, but I fully understand your comment. I think your key concern is about the presence of non-reactivated normal faults in some of the profiles, and how they relate to the large-scale analysis proposed in this paper.

Firstly, in order to maximise the impact of this paper, while not making it too long, I have decided to show the seismic profiles where evidence for fault reactivation is clear. In the Appendix are shown profiles across these same faults strands showing different geometries, and fault offsets, to stress how different is the effect of tectonic reactivation on particular structures – please, see the details in *Section 5. Kinematic Indicators of Uplift*. The paper itself concentrates on interpreting, and explaining, the presence of the clearest of reactivated faults in the study area – the 26 faults referred at the start of the paper. These 26 faults represent the principal structures that were reactivated and laterally linked to generate the long faults represented in Figures 6 and A4. Smaller faults that kept their normal offset occur at syn-rift level in the study area, but these were not

mapped as they do not offset, nor deform, the Top Cretaceous marker horizon. They were, essentially, not reactivated in a significant, discernable, magnitude.

A second important point is that, in the study area, there are multiple instances in which thrust faults change laterally into a reactivated normal fault geometry, and that is why the maps in Figure 6 and Appendix 4 are important – their symbols actually show that many of the thrust/reverse faults mapped in this paper change laterally into reactivated normal faults or are just 'passively' reactivated, using a term first stated in Terrinha et al., 2003). I think the crux in Figures 6 and A4 is in the symbols used, and I kindly invite the reviewer to verify the difference between the open/white triangles and the full, black triangles on this map. The open/white triangles, or 'dents', reflect reactivated normal faults, not thrust faults. They were chosen as symbols to highlight the important fact they were significantly reactivated.

In order to avoid any misunderstandings by the readers, I added a remark in the Caption to make clear that white triangles represent a different geometry of fault when compared to the black triangles.

**3) Vertical exaggeration is major on the seismic profiles. It is misleading not to include even an estimate of what this could be. On Fig. 5, for instance, assuming a high velocity of 4000 m/s for the whole line, we would have a vertical exaggeration of 3.3x (Fig.4 and Fig. 7 have around 1.7-1.8x exaggeration with that same velocity; App. A1 is 3.1x exaggerated; App. A2 is a whooping 4.7x exaggerated if not more!).**

Reply: Vertical exaggeration (Z-scale on Petrel) is consistently 6x on all seismic profiles. Following your advice, I have now added a sentence to all captions of seismic lines to indicate exactly this. The vertical exaggeration was not manipulated when extracting the seismic sections – only the naturally, scale-independent zooming on particular features and structures. In other words, some profiles were zoomed in on particular faults and structures.

Please, note that some seismic profiles have their tops near the 0.0s mark, while others start much deeper at c. 4.0s or more. This may be the reason why it seems that vertical section seem to have been exaggerated in different ways, though they are consistently shown with a Z-scale of 6x.

Following the remarks above, I checked all scale bars on seismic lines to be sure that their length is the correct on the paper.

**4) Naturally, vertical exaggeration depends on the velocity and since shallow velocities can be very low, then the shallower portions (water velocity 1500m/s) are highly exaggerated and the deeper ones are possibly not as exaggerated as calculated above (basement velocity can be 5-6000 m/s or higher). Petrel and other software tools provide excellent options for depth conversion. I strongly encourage the author to provide even a first-pass depth conversion (water at 1500 m/s would make a HUGE difference in how the uplift-related bathymetry is perceived) and in its defect an estimate of vertical exaggeration.**

Reply: This is an important point, but my experience tells me that converting the water column into depth will not make much difference to the overall recognition of geometric coherence in faults – particularly if the measurements of uplift and horizontal advection are taken at a shallow position in the crust, as in this case. If one changes the Z-scale on Petrel, the vertical distance on-screen will

follow this Z-scale change. Furthermore, Displacement-Length plot are often interchangeably replaced by Throw-Length plots when analysing faults. i.e. so the concepts proposed by J. Walsh and co-authors as proving the presence of coherent fault zones, or strands, can be successfully used in seismic data in the time domain.

In other words, the recognition of geometric coherence in the 26 mapped faults, in a South to North direction along their strike direction, is one of the main findings in this paper. This can be attained with data on the time domain. The measurements provided as appendices, for instance, should not be considered not as absolute values for throw and heave – or uplift and horizontal advection – but the usual time-domain data commonly used in analytical papers on rift-related faults, polygonal faults, etc.

**5) Just for the sake of demonstrating the importance of not looking at reflection seismic only in the time domain, an illustration of the lines in Fig 5 and Appendix A1 are shown below, scaled very approximately so that every second TWTT is equal to 2 km (4 km/s seismic velocity). I think the difference in how one could interpret these lines is clear. Of course, the author has traced these structures across multiple lines for consistency, and has even uncovered internal coherence in his interpretation, but I find a much better case needs to be done to convince the reader that the interpretation is actually correct. Sorry Tiago for being a pain with this: I think we really need to be extremely careful with seismic! No ill intention meant, but I would really expect to see a better clarification of why you have picked faults the way you have, and what uncertainties there may be in the interpretation and what impact (if any, of course) this would have on the rest of the analysis.**

**Reply:** It is nice to verify you have checked this important point, so apology accepted! Actually, the seismic line below was 'squashed' vertically on a drawing program and, all in all, show an interpretation similar to mine – see also my reply to Point #2 in this reply letter. The only difference is with the series of normal faults on the right of the top figure, which I interpret as a series of strike-slip reactivated normal faults on the basis of the very sharp offsets (reverse offsets) that are observed near Upper Cretaceous strata. These offsets are not common and should reflect a mixed compressional/extensional reactivation of normal faults that you indicate below. These faults should be east-dipping – in contrast with your interpretation – as shown by parallel lines to the North and South of the seismic lines shown.

Again, a detailed and correct interpretation of the maps in Figure 6 and A4, combined with more detailed text in their captions, will make the seismic lines easier to understand.

An important point in your figure below is that you have emphasised the presence of normal faults in the syn-rift succession, but the evidence for reactivation should be gathered further up in the succession, at the top Mesozoic and Cenozoic levels, where folded and offset strata are present. Any gentle folding in the syn-rift succession may well be related to differential compaction above 'harder' footwall tips, so some of the normal faults you interpret (in your figure) below the top Cretaceous marker were not part of the 26 faults considered as having been significantly reactivated, and mapped, in this work. The thrust fault in the section shown in your review, and Fault 3 (the second of the normal faults indicated, going from NW to SE) have been mapped in this work as reactivated structures, but not the other normal faults you mark in the figure. They were left uninterpreted because there is no apparent reactivation affecting them.

In summary, there are non-reactivated faults in the study area that kept their syn-rift normal geometry throughout the late Mesozoic and Cenozoic. These were not mapped as reactivated faults in Figure 6, and throughout this work. In essence, the 26 faults mapped in Figure 6 and A4 are the ones demonstrating significant, unequivocal reactivation, with many other faults having been kept with their syn-rift geometry during tectonic compression, and thus demonstrating very little (if any) tectonic reactivation. These non-reactivated faults are also not laterally linked as the reactivated ones are.

[Figure]

**6. The same applies to the analysis of throw and heave (Section 6). You are combining different magnitudes (time and true distance). It is well worth clarifying this and stating any possible caveats.**

**Reply:** As with the 'classical' D-L and T-Z plots for faults, I do not feel there is a problem in plotting distance and data in the time domain. D-L (displacement-length) plots can, and are, usually changed into Throw-Length plots when Vp velocities are not 100% sure in a specific area, and borehole data is sparse or unavailable. T-Z (throw-length) plots are used systematically to understand the vertical distribution of throws (in time) along the height of a fault. In practice, there is no problem of using time-domain data and plot it against true lengths of faults as a way of demonstrating coherency in fault growth and propagation – it is the shape of the T-D and T-Z curves that are important in such analysis, not absolute values of throw, whose variation should be minimal if measurements are gathered at a similar stratigraphic level – as in the case of this paper.

Nevertheless, I have converted all ms values into metres by using the 'common' Vp value of 2000 m/s in all conversions. Figure 8 and the graphs in Figures 9 and 10 were changed to metres when considering (and plotting) fault offsets, or uplift.

**7. On the line in Fig. 7, the depth in time goes down to 11s TWTT. This depth is huge! Most likely, some of the reflectors you are seeing could even be Moho!!! I know the Estremadura spur is likely not in isostatic balance (held up by far field stresses, which only makes things worse, actually), but in isostatic balance, your Moho reflections tend to come in at depths of 9-11s TWTT!! See the paper of Warner (1987, https://doi.org/10.1111/j.1365-246X.1987.tb06651.x). I encourage you to have a look at this line again and see if there is something even more spectacular that you may have missed!!! :)**

**Reply:** Very good comment, demonstrating a concern of the reviewer with the significance of the deeper structures imaged in seismic. However, I think there is some discrepancy in your comment when referring to the 9-11s TWT depth of the Moho observed in several distal regions of the West Iberian margin, particularly in its 'Ocean-Continent Transition Zone' (OCTZ), and the relative location of the seismic lines of this paper.

A first aspect is that, in the particular case of Figure 4, the seismic reflections identified at c. 11s TWT are deep, but the seafloor is also at 6.0s TWT. This is a depth difference of c. 5.0s, which is not that great for a proximal/intermediate part of the continental margin where the lithosphere is relatively thick. A second important aspect is that the seismic line shown is located quite proximally (i.e. landward) on the continental margin. So, the reply is: Yes – the Moho appears at a depth of c. 9-11s TWT in the more distal parts of West Iberia, namely in the regions interpreted as part of the OCTZ, but occurs much deeper near the continental slope, on the more proximal margin, as in the case shown in Figure 4. This Moho reflection is usually not imaged on 2D or 3D seismic reflection data, near the continental slope and shelf, due to its greater depth.

Please, see the figure below from Afilhado et al. (2008) https://doi.org/10.1016/j.tecto.2008.03.002 showing an interpreted refraction profile across SW Iberia - the projected position of Figure 4 along this transect is in the more proximal part of the ThD zone (Afilhado et al., 2008). In this region of the West Iberian margin, the Moho is at a depth of 28-29 km.

[Figure]

**8. Fault 3: I understand this becomes a key structure in your analysis. I have, however, a question as to whether this is really a single fault, and as to whether it really is an extensional fault. On Fig. 4 you clearly identify it as a SE dipping thrust. Why do you not interpret it the same way on Appendix A1? I think it would make much more sense! In any case, if you interpret the fault to change from extensional (transtensional?) to thrust along strike, then you shouldn't really interpret it as being linked. If you observe the throw map for this fault (Fig. 8a, 9a) it seems clear to me that it could actually be formed by three independent segments. As you are working on 2D seismic, I would like to see a better argumentation in favour of this being a single, long (linked-up?) fault.**

**Reply:** Yes, that is exactly the point of this paper: there were rift-related faults, and distinct fault segments or strands formed during tectonic extension, that are now laterally linked to accommodate tectonic compression. Not all the rift-related faults have been reactivated – only the ones shown in Figures 6 and A4 as forming large linked fault zones, or networks as defined by Peacock (2000) – Glossary of Normal Faults - https://doi.org/10.1016/S0191-8141(00)80102-9. Most of these faults were reactivated from their 'normal' geometry to form, at present, reverse faults – which are of relative high angle put reflect the 'passive' reactivation of older normal, rift-related faults – and thrusts intersecting older extensional structures or rooted in them, at depth.  If reactivated in unison, as a single fault zone, some of these linked fault networks may generate earthquakes of a magnitude of Richer 8.0.

Nevertheless, I changed the interpretation in Figure 4 to stress the presence of reactivated normal faults on the seismic profile provided, as per Figure 6. Indeed, Figure 6 was not correlating well with Figure 4.

**9. I am sorely missing a structural map (i.e. isobath or isochron map) of a pre-uplift horizon. The map in Fig. 6 is misleading because the fault cutoffs shown are not referenced to any specific interpretation horizon. The Base of Miocene is a likely candidate. The reason that I ask this is because the structural map is a very quick way of visually assessing the relative throw on each fault. I think this would be a major positive contribution to the article – this helps remove ambiguity on your fault interpretation in a very significant way.**

**Reply:** I agree with this comment. Lack of time made it hard to review (and improve) the Petrel maps I already have compiled in particular parts of SW Iberia. I have now compiled twt-structure maps of the Top Cretaceous and Basement horizons taken from Petrel, as shown in the new Figure 7. However, on the interpreted Petrel project it is virtually impossible to work on 'pre-uplift' horizon without completing a thorough, time-consuming, reconstruction of the pre-inversion geometry of faults and basins. This can be done on another paper, but not in this (already-long) article.

**10. There is a major argumentation flaw (in my opinion) in applying some of the criteria in section 3.2 further down. In Section 5.2 you talk about folded strata and you previously mentioned in section 3.2 that stratal thickness is an important criterion in distinguishing post-rift from syn-rift fault activity. However, your seismic images are shown in time, with partially extreme vertical exaggeration (the values I calculated above are low range values for your sediments, that likely have seismic velocities lower than 4km/s and are therefore more vertically exaggerated). You can only use this criterion if you somehow attempt to display these same structures in depth. In my view, as this is, it is not admissible (sorry for the harshness, no ill intention).**

**Reply:** There is no extreme vertical exaggeration in the profiles (see point 3) and the distinction between syn- and post-rift strata is straight forward by looking at their distinct geometries, let alone because there is a breakup unconformity, and overlying sequence, marking the end of the syn-rift strata. Please, see Soares et al. (2012) EPSL, Alves and Cunha (2018) EPSL, and Alves et al. (2022) Marine and Petroleum Geology. Hundreds, if not thousands, of papers on rift-related tectonics and associated strata have been published thus far using data on the time-domain so my feeling is that, in your comment, you were expecting these 'growth strata' to be rather thin. In fact, I refer in my paper to large growth packages that relate to the scale seismic-stratigraphic sequences or even megasequences. That is precisely the reason why I chose two unequivocal stratigraphic markers to measure uplift and horizontal advection – one regional unconformity at the top of the Cretaceous, known to coincide in other parts of Europe with the start of the Alpine Orogeny, and a mid-Miocene unconformity that is prevalent on the margin and was previously interpreted by authors such as Gracia et al. (1998), Terrinha et al. (2003) and more recently Somoza et al. (2022) as being a pervasive seismic-stratigraphic marker in SW Iberia. That is also the reason why I discarded the analysis of any smaller normal faults that show no major reactivation above the Top Cretaceous marker – one cannot distinguish between the contribution of local differential compaction and minor tectonic reactivation over the footwall tips that do not deform the Top Cretaceous horizons.

I stress once again that my objective is not to 'nit-pick' thin seismic-stratigraphic packages using regional 2D seismic data; it is to concentrate on the large-scale coherence of faults under tectonic reactivation using regional seismic markers. Hence, the approach taken in my mapping was that

stratal growth and fault reactivation ought to be clear to the interpreters, and of a relatively large scale, when using regional 2D seismic profiles. The smaller-scale analysis of strata growth in small packages, or units, can be made using 3D seismic data of high resolution, for instance, but the method described in this paper should be applied at a smaller scale of analysis. Unless the interpreter has access to large mega-merge 3D data covering the entire SW Iberian margin.

**11. Salt tectonics is prevalent both to the north and to the south (Pena dos Reis et al. 2017, https://doi.org/10.1016/B978-0-12-809417-4.00015-X; Casacao et al. 2023, http://dx.doi.org/10.1306/08072221100; Matias et al. 2011, http://dx.doi.org/10.1306/01271110032; Ramos et al. 2017a, https://doi.org/10.1016/j.marpetgeo.2017.09.028) but is not interpreted here. I understand it has been previously stated that this basin is peculiar because of this absence of salt tectonism. However, on the seismic data provided, there are isolated features that may be diapirs based on their post-rift and pre-uplift growth strata and short wavelength. Most evident may be the structure under the text "blind thrust" above Fault 25 on Fig. 4b. It is somewhat reminiscent of the wavelength of structures in the westernmost Algarve basin (cf. Fig. 12 in Ramos et al. 2017a). Furthermore, even if the Dagorda salt was not a player in syn-rift salt tectonics, it would most likely have acted as a detachment during shortening. Indeed, your "thrust-and-fold-belt geometries" are somewhat reminiscent of fold belts with a well expressed detachment.**

**Reply:** Key papers missing in the list above are Alves et al. (2003) MPG, Alves et al. (2009) Tectonics and Ricardo Pereira's three papers on the SW Iberian margin and its structure. Alves et al. (2006) was the first analysis of deep-offshore seismic profiles where salt was identified in the Peniche Basin, while my own PhD thesis was about salt vs. fault-related tectonics in the Lusitanian Basin. Pereira et al. have concluded on the structural framework and evolution of SW Iberia in the context of Atlantic rifting and Africa's movement.

The fact of the matter is that there is very little, if any, salt in SW Iberia, though in the past I have recognised local 'rafting' of the syn-rift succession at the very SW tip of the margin in industry 3D seismic data. The main reason for this lack of Uppermost Triassic-Lower Cretacous salt may relate to the margin's thermal/magmatic history, as there is an Early Jurassic phase of CAMP-related volcanism in this region. This volcanism may have kept the SW Iberian margin slightly uplifted in relation to the Algarve, Lusitanian and Peniche Basins. Such early volcanism (syn-stretching sensu Rowan et al., 2014) is documented onshore near Sesimbra and Santiago do Cacem, and also in Western Algarve.

The features you suggest as diapirs are well resolved in the older GIS-84 data. They are two small folds (and related high-angle thrust faults) that occur half-way on the continental shelf. They are inversion related structures. The GIS-84 data were processed in a different way, using a different fold, and resolves these two folds quite nicely. I think I even showed them in my PhD thesis, back in 2002. They are not salt structures.

[Figure]

**12. In terms of terminology, I understand the use uplift and advection for rock masses, but when you are talking of displacement on faults, the terms that are normally used are throw and heave. Is there a reason not to use these? Furthermore, when you give values for throw and heave, you should refer them to the specific marker/horizon/stratigraphic interval for which you have measured the offset.**

**Reply:** I am using the terms 'uplift' and 'horizontal advection' based on Figure 3 and the concepts of He et al. (2022) for tectonically inverted regions. Figure 3 describes the relationship between uplift and horizontal advection. The values given in the graphs are 'maximum uplift' and 'horizontal advection' measured in between the two stratigraphic markers used in this paper – The top Cretaceous horizon and the Top Miocene horizon.

I change the wording in the paper to take this aspect into account. Please, see Lines 229 and 230.

**Minor comments:**

**line 50 – I understand it lies "around the corner" but a comparison to the well documented example of Ramos et al. 2017b (https://doi.org/10.1002/2016TC004262) could be interesting** – Thank you for this remark. This reference was added to the paper. New detail was also added in the text wherever Ramos et al. (2017) is cited.

**line 91 – "in what ways do tectonic"** – Verb 'do' was added to this research question.

**line 92 – "and how do they interact"** - Verb 'do' was added to this research question.

**line 101 – please check the recent publication of Amigo Marx et al. 2022 (https://doi.org/10.1111/ter.12595)** - Reference was added to the paper.

**line 146 – there is an event of re-activation of salt structures in the Algarve basin in the Early Cretaceous and related sedimentary record (Ramos et al. 2016, https://doi.org/10.1007/s00531-015-1280-1) that likely relates to the regional picture (note that** – Yes, but the start of Iberia's rotation and tectonic compression post-dates the Early Cretaceous. Reading Ramos et al. (2016), giving emphasis to their Figure 2, it is obvious that the period in which inversion started in SW Iberia, for which there is a tectono-stratigraphic record, is stratigraphically missing in Algarve. Algarve records exhumation and non-deposition at this time. Early Cretaceous tectonics, in SW Iberia, is included in the syn-rift succession, and has been explained by Alves and Cunha (2018) and Alves et al. (2022) as syn-breakup.

**line 155 – inversion in the Algarve extends beyond the Miocene (Ramos et al. 2016, 2017b)** – These two references, and a new sentence, were added to the paper. Please, see the instances in which Ramos et al. (2016) is cited in the text.

**line 183 – He et al. missing year –** Corrected.

**line 187-193 – slight repetition** – Corrected.

**line 197-198 – beds are not really deposited horizontally on continental margins (the syn-uplift units on your own seismic betray you) ;) I think it is worth stating it is near-horizontal (my personal battle against assuming pre-deformation = horizontal)** – Point taken and sentence was changed.

**line 217 – the faults on Fig. 6 look like they are NE-SW striking, not NW-SE. Can you please check this is correct? Also, if your faults indeed strike NW-SE then sections in Fig. 4 and Fig. 5 would be parallel to your fault, which I think is not what you interpret? –** This orientation was checked and changed. 'NNE-SSW' was changed to NE-SW.

**line 219 – "on seismic data" –** This is actually a common error in papers compiled by non-native English speakers…that I used to commit too. It should be 'in' seismic data as the subject here is 'data', i.e. seen in that particular data. It is, however, 'on' the seismic profile.

**line 228-29 – NW-dipping normal faults reactivated as SE-dipping reverse faults is a pretty paradoxical statement... I understand it, but I think it would be useful to discuss this style of inversion (compare to the very similar style of inversion described by Ramos et al. 2017b for the Guadalquivir and Portimao Banks in Algarve). As in the example of Algarve, I think this speaks for an inherited basement fabric and could speak for both basins sitting on a similar SPZ-style basement (with inherited Variscan thrust fabric that re-activates as opposed to inversion of normal faults). (?) –** This part was significantly changed, if not mostly deleted, after Reviewer #2 provided their comments.

**line 267 – to speak of 100s of meters, you should provide a vertical scale in meters. –** This comment refers to the Lusitanian Basin, and parts of the Estremadura Spur, as the Cenomanian marker is not apparent in seismic data south of Sesimbra/Arrábida. I did provide, nonetheless, a conversion to seconds TWT in the text – see the end of sub-section 3.1.

**Fig. 6 – can you please check Appendix B and Fig. 4 are correctly placed here? There is a line labelled SF2 which seems to be App B, and App B seems to be Fig. 4? If Fig. 4 is not on this map, please locate it. –** checked and changed. Please, see the new Figures 6 and Appendix B.

**Fig. 6 – Fault 3 on Fig. 4 is a SE-dipping thrust, but on Fig. 6 it is represented as a NW-dipping thrust/inverted fault. –** Yes. Please, see the reply to Item #8 in this reply letter.

**Section 6 – the analysis here is mixing depth in TWT milliseconds with horizontal distance in meters. This requires some very careful clarification. –** Please, see reply to Item # 6 on the previous pages.

**line 320 – provide a Fig. reference for faults 3,7,11. (Fig. 5?).** Done.

**line 329 – sorry, not sure where you see that the intrusions drive inversion? Did you really explain it before? Wasn't clear to me at least. OK, is it what you later say in line 340? Then you need a reference here – it is not something you are demonstrating with your data. –** No. I never state that intrusions drive inversion in this paper. I say they form buttresses to the NW-SE directed compression SW Iberia has been subjected to, up to the present day.**line 340 and further – if igneous bodies are so important in your inversion history, it is worthwhile including a seismic line that shows these relationships. –** Not the igneous bodies per se, but the presence of a hard, intruded crust that formed a buttress to tectonic compression. Where there is a clear influence of magmatic bodies on local uplift – I stress local – is documented further North by Pereira and Gamboa (2023) – Geology (https://pubs.geoscienceworld.org/gsa/geology/article/51/9/803/623410/In-situ-carbon-storage-potential-in-a-buried). These authors had access to 3D Seismic Data and mapped the magmatic bodies in great detail. Though present in our study area, we cannot map such features as they are present to the North of the mapped region (Fontanelas Volcano) and out of the span of our 2D seismic grid (Estremadura Spur Intrusion).

**lines 357-358 – deformation is really mostly accommodated by the folds on the left part of Fig. 7. The thrusts that you interpret linking to the extensional faults are really minor. They don't really accommodate much shortening... furthermore, I am not fully convinced that what you are seeing at the base of the thrusts is really syn-rift strata and not some form of deep crustal reflectors. –** They are not deep reflectors as there is a rough W-E seismic line crossing this one with developed syn-rift strata at this same depth. This E-W seismic line is Figure 7 in Alves and Cunha (2018) - https://doi.org/10.1016/j.epsl.2017.11.054

Please, see also my reply to Item #7 on the previous pages.

**line 390 – I don't fully follow this argument. Thermal subsidence is well known for passive margins. I do not know that it applies to inversion setting (!?) nor to intrusions (whose thermal-related dilation potential is probable limited). –** This argument has to do with the fact that the Estremadura Spur was inverted, and uplifted, since the end of the Cretaceous, but that its regional 'pop-up' style of deformation was certainly combined with thermal cooling of 'transitional' and oceanic crust in the Tagus and Iberia Abyssal plains to generate the 4000+ m of bathymetric difference between the Spur and its Northern and Southern flanks. It will be virtually impossible to discern between the tectonic-uplift component (on the Spur) and the thermal-subsidence contribution (on the abyssal plains) using seismic data alone. Therefore, any uplift, horizontal advection and throw measurements along faults flanking the Estremadura Spur will return values that are significantly (if not strikingly) larger than any other faults in West Iberia. Therefore, such measurements were discarded in this work.

**Fig. 8 – you place your throw and heave bars always on the western side of the faults. However, I find this misleading and I think it would be much more accurate to place them always in the hangingwall or always in the footwall, or any other placement that makes kinematic sense (block up vs block down). But placing them always west seems to indicate these are the upthrown or downthrown blocks consistently, which is not the case. –** Actually, this is the case in the majority of faults; bars are located in their downthrown blocks. To plot the bars as you suggest was first thought when compiling the paper, but it was seen to result in overlapping bars and text when faults are closely spaced.

**Fig. 9 – you should NOT plot depth vs time domain units. This is not valid. Please revise, provide an approximate conversion factor or something similar before you attempt this. –** I can use the usual, average Vp velocity of 2000 m/s, but the values and relation shown will be exactly the same. The TWT distance between two reflectors, or seismic markers (i.e., the 'uplift' or 'throw' values) when measured on seismic data may vary depending on seismic velocities, but if these measurements are consistently made within a relatively thick interval, with a Vp velocity that is relatively constant on average, the variations will be minor and not very important at this scale of analysis. They will be only important if I start measuring uplift and horizontal advection in, let's say, Miocene strata along a fault, to then continue to do the same in 'hard', high-Vp Lower Jurassic strata, or in Palaeozoic basement units. Velocity profiles extracted from the seismic profiles would show very little Vp variations, or gradients, at the depths in which uplift and horizontal advection were measured.

Nevertheless, I take the comment on board and converted ms into metres in the graph, and in other maps in the article, using a Vp value of 2000 m/s that was taken from wells Go-1 and Pe-1 for Cretaceous sediments. Please, see the end of Section 2. Data and Methods.

**Fig. 9, 10 – it is not clear what your horizontal axis is. It needs to be clarified and made evident in a location map. I assume you are referring to dip seismic profiles? Are they parallel to your assumed**

**shortening direction? If not, then you cannot really add up displacements... You need to state these assumptions better. –** Assumptions stated in the caption. Please, see the new captions.

**line 430 – this is an interesting thought. However, shortening should be accommodated along strike equally. What happens to the shortening that is not expressed around the intrusions? Is this internal strain or is it being transferred elsewhere? –** No. Strain is known to preferentially concentrate – in geological time and space (4D) – near the centre of faults, or fault zones, that are coherent in their kinematics. This occurs for any tectonic setting. Please, see Walsh et al. (2003), for instance.

**lines 459-463 – this is very interesting point and perhaps is worth bringing into the discussion. After all, this provides a mechanism to accommodate the type of dynamic topography implied by the lithospheric folding model (Cloetingh et al. 2002, https://doi.org/10.1029/2001TC901031) –** Maybe, but Sierd Cloething et al. paper above refers to lithospheric folding of Iberia at the largest of scales, i.e. at a scale that is larger than the study area in this paper. I am not saying the study area is not large enough, but Cloething et al. (2002) relate the presence, in Iberia, of lithospheric folds that are of the wavelength of the Variscan lithosphere. In contrast, I feel the phenomena we are recording are constrained geologically by the syn-rift structural fabric of the margin and subsequent magmatic and tectonic shortening recorded after the Late Cretaceous. In the study area, there may be a Variscan control on the rift-related fabric developed in the Mesozoic, but while Cloething et al. (2002) refers to fold amplitudes of ~200 km (see their Figures 8 and 11), the folding and deformation styles recorded in this paper is of the scale of a dozens of kilometres. Because the reactivation of syn-rift faults is the predominant phenomenon recorded in our study area, the wavelength of such a deformation follows the fabric of grabens and half-grabens below – and their bounding faults – which are typically spaced by 15-20 km.

**line 465 – I do not see that the authors have documented intrusion-related uplift... Sounds like a conjecture? Please check this is adequately argued. –** Not a conjecture; uplift near igneous intrusions is a universal phenomenon documented, for instance, by C. Magee et al. (2017) https://doi.org/10.1130/G38839.1, Roelofse et al. (2020) https://doi.org/10.1111/bre.12507 or Jing et al. (2022) https://doi.org/10.1016/j.margeo.2022.106933

Trusting these changes will justify the publication of this work in EGU's Solid Earth.

Prof. Tiago M. Alves

(3D Seismic Lab – Cardiff University)